# Evaluation of sea turtle morbidity and mortality within the Indian Ocean from 12 years of data shows high prevalence of ghost net entanglement

**Katrina Himpson***, **Simon Dixon**, **Thomas Le Berre**

Reefscapers Ltd Plc, Malé, Republic of the Maldives

* mdcmanager.mlg@fourseasons.com

**Data Availability Statement:** All relevant data are within the manuscript and its Supporting Information files.

## Abstract

Anthropogenic activities can negatively affect sea turtle populations. Quantifying the effect of human actions on these threatened species can help guide management strategies to reduce adverse impacts. However, such assessments require extensive effort and resources and as such have not been carried out in many areas of important sea turtle habitat, including the Republic of the Maldives (Maldives). Here, we utilise 12 years of data (2010–2022) collected from marine turtle stranding and rehabilitation cases from across the Maldives to identify the key threats in this region. Olive ridley turtles were found stranded or injured most frequently (74.7% of total cases), along with hawksbill (15.2%), and green (10.1%) turtles. Anthropogenic factors were the primary cause of injury or stranding in 75.2% of cases with entanglement in ghost fishing gear being the most common (66.2% of all cases). Other causes of morbidity, such as from turtles being kept as pets (5.6%), boat strikes (<1%), bycatch (<1%), and poaching (<1%) were recorded less frequently. Olive ridley turtles were more likely to have injuries associated with entanglement than other species and showed a peak in admissions during the northeast monsoon, in the period following the known arribada nesting season in nearby India. Turtles admitted to rehabilitation following entanglement were released a mean of 70 days sooner and had 27.5% lower mortality rates than for other causes of admission. This study highlights the high prevalence of ghost net entanglement of sea turtles within the Maldives. The topic of ghost fishing is of global importance and international cooperation is critical in tackling this growing issue.

## Introduction

Human activities have substantial impacts on the worlds' oceans and the species which live in them [1]. Anthropogenic factors such as overexploitation, habitat loss, climate change, invasive species, disease, and pollution can negatively affect wildlife populations and contribute to species declines or extinctions [1,2]. The impacts of these activities are more pronounced in large bodied species which are subject to more intense pressures, for example through

**Funding:** The author(s) received no specific funding for this work.

**Competing interests:** The authors have declared that no competing interests exist.

overexploitation, and are vulnerable to extinction due to slow life histories [3,4]. As marine megafauna convey a variety of environmental, economic, cultural and social benefits disproportionate to the overall percentage of species they represent, and can additionally act as umbrella species for conservation, they should be considered a priority for protection [3,5].

Marine turtles are one group of marine megafauna under threat of extinction through human activities; primarily through interactions with the fishing industry, overexploitation, and marine pollution [6–8]. Although six of the seven species of marine turtle are considered to be under threat of extinction, through extensive conservation efforts populations are stable or increasing in many regions [9–11]. However, as the type and magnitude of threats to marine turtles varies between geographic regions, it is important to consider that management decisions to mitigate anthropogenic impacts in one location may not be effective in another [12–14]. To safeguard against future losses and facilitate further population recovery it is critical to identify and quantify threats to marine turtles on a regional scale.

Regardless of the importance of assessing threats to marine turtles, the process remains challenging: all species are elusive with pelagic life stages, making gathering the large datasets required for accurate evaluations labour intensive and costly [15]. Given the extensive resources required to assess threats, these have been performed only within certain well-studied populations; namely of green turtles in the Americas and Australia, and loggerhead turtles in the Mediterranean [8,12,16–18]. However, threats to marine turtles remain unassessed in many regions, including areas with significant populations [15].

The Republic of the Maldives (Maldives) is one region of important marine turtle habitat where a comprehensive evaluation of threats has not been conducted [11]. However, a rapidly expanding and increasingly environmentally focused tourism industry over the past few decades has facilitated the collection of comprehensive and long-term data across many areas of marine science in the Maldives. [19,20]. Here, we utilise data collected from stranded turtles and those admitted into rehabilitation centres to evaluate the threats to marine turtles in this region.

Stranding data is a common method of assessing causes of morbidity and mortality in marine turtles [8,12,21]. Although stranded turtles found on beaches or floating on the ocean's surface only represent a small proportion of total deaths and injuries; strandings are considered representative of threats and allow estimations of the scale of local hazards to be made [14,21]. Where stranded individuals are found alive and admitted to rehabilitation centres, longitudinal observations made on progress and recovery can provide additional data towards a more comprehensive overview of threats to marine turtle populations in a region [22].

Five of the seven globally recognised species of marine turtle have been recorded in the Maldives. Green (*Chelonia mydas*) and hawksbill (*Eretmochelys imbricata*) turtles are permanent residents and are sighted frequently throughout the region [23]. Both species hold neritic foraging grounds which are established after an initial pelagic life-stage as young juveniles. Nesting is reported in several atolls with animals known to migrate from the Chagos archipelago; indicating that the Maldives provides important nesting habitat for turtles in the region [24–27]. A recent regional IUCN evaluation has classified hawksbill turtles as 'critically endangered' and green turtles as 'endangered', matching global assessments, although a recent evaluation suggests that populations in the area are stable [11,28].

Olive ridley turtles are found more frequently in pelagic habitats than neritic and are known to have large nesting populations along the east coast of India [29]. In the Maldives they are most frequently sighted offshore and have no known resident or nesting populations [23]. However, olive ridley are found entangled in ghost nets;fishing nets which have been lost or discarded, with relatively high frequency within the atolls, particularly during the northeast monsoon (January to March) where mass nesting (also known as arribada behaviour) along

the east coast of India overlaps with a peak in trawl fishing in the same area [30–32]. The strong monsoon currents then wash injured and entangled turtles into the Maldives [30–32].

Although both loggerhead and leatherback turtles have been reported within the Maldives, both species are infrequent and transient visitors with no known resident populations [23].

This study represents the first long-term, multi-species analysis of sea turtle morbidities and mortalities in this area of the Indian Ocean. Using 12 years of stranding and rehabilitation data collected within the Maldives we aim to: analyse initial status and cause of injury in stranded animals, compare these between species and life stages, determine overall mortality rate of animals found alive, and identify seasonal patterns in strandings.

## Methods

### Background

The Maldives is a chain of coral atolls running along a north-south axis around 400km to the southwest of India (07˚06'N—00˚41'S, 72˚32'E—73˚45'E) (Fig 1). The climate of the Maldives is tropical and has 2 distinct seasons: a hot dry period during the northeast monsoon (January to March), and a hot rainy season during the southwest monsoon (July to September) [33]. Although the Maldives exclusive economic zone (EEZ) covers an area of over $90,000 km^2$, only 0.3% of this is above sea level, principally as small, low-lying islands [34]. Arranged in 26 geographic atolls, the 1192 islands of the Maldives are mostly undeveloped with only 194 islands inhabited and a further 150 developed as tourist resorts The remainder are undeveloped or are used for industries such as agriculture [35]. Tourism, which was initiated in the early 1970's, has rapidly expanded to become the largest economic sector, driven by the tropical climate and high diversity of marine life which attracts high numbers of international visitors every year [35]. Both human population densities and visitor numbers are higher around the more accessible central atolls [36]. Surrounding the coral-built islands and covering around $8900 km^2$ the shallow reefs of the Maldives are the 7th largest and 5th most biodiverse in the world [37]. This system supports a broad diversity of species, including numerous which are endemic, rare or threatened species including corals, elasmobranchs, cetaceans, and marine turtles [35].

The Maldives Sea Turtle Conservation Program (MSTCP) was established in 2010 to support threatened marine turtle populations in the Maldives. The program is a collaboration

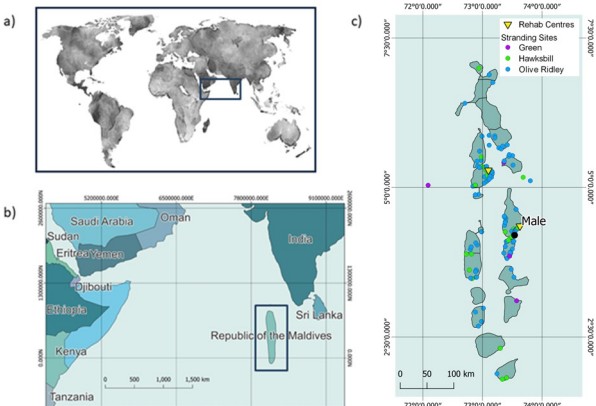

**Fig 1.** (a) Location of the Republic of the Maldives on a global scale, (b) within the Indian Ocean and (c) the geographic structure of the Maldives showing the double chain of atolls, locations of strandings recorded between 2010 and 2022 and the 2 rehabilitation centres.

between marine consultancy company Reefscapers Ltd and Four Seasons Hotels and Resorts and operates under Environmental Protection Agency (EPA) Protected Species Research Permit number EPA/2020/PSR/T02. The MSTCP conducts sea turtle rehabilitation and research in 2 locations: Landaa Giraavaru in Baa atoll and Kuda Huraa in Male (Kaafu) atoll (Fig 1).

## Data collection and processing

Data was recorded from injured or stranded turtles which were reported to the MSTCP between March 2010 and September 2022. Animal handling and husbandry practices during this process followed recommended and best practice sea turtle care and management guidelines [38–41]. Turtles reported to the MSTCP had a standard set of information recorded: species, curved carapace length (CCL), life stage, sex, date found, identity of the reporting party, initial status, cause of injury or mortality, details of injuries or abnormalities present, final outcome, and date of final outcome. This information, excluding data pertaining to the final outcome of the case, was recorded on admission for rehabilitation. In cases where admission did not occur data was recorded from verbal descriptions and visual media (photos and videos) provided by the discoverers of the turtle.

Life stages were categorised as 'pelagic-stage juvenile', 'neritic-stage juvenile', and 'adult' for green and hawksbill turtles. Life stage was determined by CCL with 30cm used as the delineation between the 2 juvenile stages. In the Maldives green and hawksbill turtles are seen to return to neritic feeding grounds from approximately 30cm in length [23]. Green turtles with CCL greater than 95cm (males) and 100cm (females) were classed as adults [42], with 75cm used as the cut-off for hawksbill [43]. The life stages of olive ridley turtles were divided into 'juveniles' and 'adults' as all life stages are primarily pelagic in nature, with 60cm CCL used as the divide between groups [44,45]. As adult turtles are sexually dimorphic, sex was determined in mature individuals using distinctive physical characteristics such as tail length. No attempt was made to determine sex in juvenile animals.

Initial status was defined as the state in which the stranded or injured turtle was found. Cases were divided into one of 5 categories; entangled (turtles ensnared in marine debris such as ghost nets, ropes, or other anthropogenic debris), beached (debilitated animals found on the shoreline), floating (those found on the oceans' surface but not entangled), kept as a pet (animals previously held in captivity and subsequently seized or surrendered), and unknown (where the initial status had not been recorded).

Cause of injury or mortality was determined by several means: initial status, clinical examination, and post-mortem examination of deceased individuals. Injuries and abnormalities were described and categorised into likely causes using previously published descriptions of gross lesions caused by different means [6,22]. For example, linear lacerations to the proximal limbs and neck as well as linear abrasions to the carapace or plastron were considered characteristic of entanglement in netting or analogous materials. Similarly, parallel linear damage to the carapace, or less frequently the plastron, associated with severe internal trauma was attributed to propellor injury from a boat strike. The discovery of a complete carapace or plastron with toolmarks was considered to be indicative of poaching. Injuries were classified as abrasions (surface damage to the skin or shell not involving deeper tissues), lacerations (more severe damage to soft tissues involving underlying muscle and connective tissue), fractures (broken bone), missing (previous traumatic amputation of a limb), and carapacial damage.

Causes of morbidity were then categorised as natural or anthropogenic in origin. Natural causes included infection or cachexia (emaciation with no discernible primary cause, as determined by Body Condition Index (BCI) [46]), whilst anthropogenic causes of injury incorporated entanglement, boat strikes, hook injuries, and, for animals kept as pets, poor husbandry.

The difference between date found and date of final outcome, i.e., when the turtle was released or died, gave the time taken to reach an outcome in days.

To assess seasonal differences in frequency of stranding reports the year was split into four seasons: northeast monsoon (January and March), first inter-monsoon period (April to June), southwest monsoon (July- September), and the second inter-monsoon period (October-December) [47].

Statistical analysis were carried out in R version 2022.07.0 using the 'epitools' package [48]. Odds ratios were used to assess relative probability of turtles being found in the various initial statuses. Chi-squared tests were used to test for variation in mean mortality rates, season of admission, and time spent in rehabilitation between groups. A 2-sample t-test was used to determine whether the time taken to reach an outcome varied between turtles which died and those that were released.

## Results

In total, 459 turtles were reported as stranded or injured between 2010 and 2022 from 18 different atolls with the central, more densely populated atolls (Male, Baa, Ari) over-represented (Fig 1). Most turtles were found alive (86.7%, n = 398) and the majority of these were admitted into rehabilitation centres (80.1%, n = 319) with others being released immediately (19.9%, n = 79). Olive ridley turtles were the most frequently recorded species (n = 343), followed by hawksbill, (n = 70) then green turtles (n = 46). No loggerhead or leatherback turtles were recorded (Fig 1).

Juvenile turtles were more commonly reported than adults with a juvenile: adult ratio of 2.5:1 in olive ridley, 14:1 in green turtles, and no adult hawksbill being recorded. This was reflected in mean curved carapace lengths (CCL) of 51.28cm (SD 12.44, n = 275), 41.03cm (SD 11.65 n = 56), and 24.98cm (SD 25.89, n = 39) for olive ridley, hawksbill, and green turtles respectively (Fig 2).

The adult sex ratio, in turtles where this was recorded, was female biased in olive ridley and green turtles with a female: male ratio of 2.75:1 and 2:1 respectively. As no adult hawksbill turtles were recorded it was not possible to determine an adult sex ratio for this species (Fig 3A).

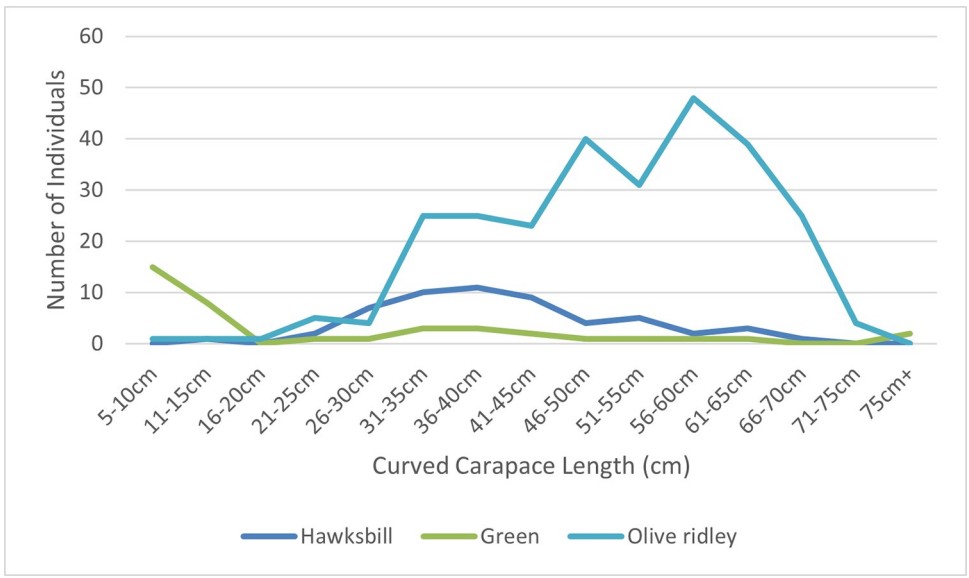

**Fig 2. Size distribution of turtles admitted into rehabilitation or found stranded within the study period.**

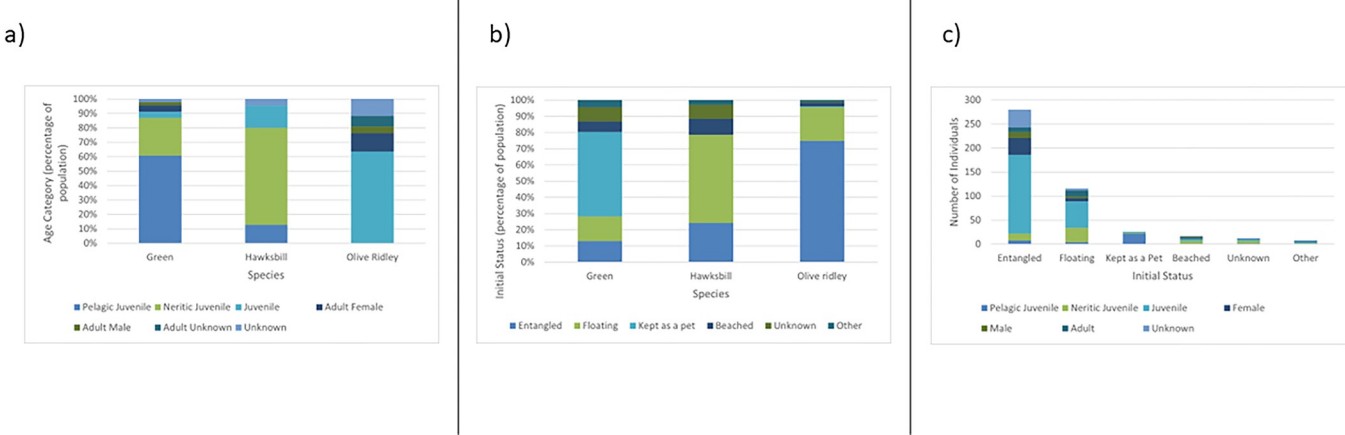

**Fig 3.** (a) Proportions of the sex and life stage of marine turtle species found injured or stranded within the Maldives between 2010 and 2022, (b) proportion of the initial status of animals between species and (c) pount of initial status between life stages.

Tourist resorts were responsible for reporting 84.1% of turtles (n = 269/320) with smaller proportions originating from non-government organisations (5.0%, n = 16), scuba diving outfits (4.0%, n = 13), local stakeholders, (4.0%, n = 13) and other miscellaneous sources (2.8%, n = 9).

## Initial status

Turtles were most frequently found whilst entangled in ghost nets (61.0% of all reports, n = 280) or floating (25.0%, n = 115). Others were reported after having been illegally kept as pets (5.6%, n = 26), were found beached (3.9%, n = 18), after being struck by boats (1.1%), poached (0.4%, n = 2), fished unintentionally (0.2%, n = 1), or did not have their initial status recorded (2.6%, n = 12) (Fig 3B).

However, the initial status of injured or stranded turtles varied between species, with olive ridley turtles more likely to be found entangled than other species (OR = 19.04; 95%CI: 8.34–52.09; p = <0.0001), hawksbill more likely to be found floating (OR = 9.67; 95%CI: 2.78–18.72; p = <0.0001) and green turtles were more likely to be kept as pets (OR = 64.96; 95%CI: 12.60–1603.45; p = <0.0001) (Fig 3B).

Different life stages were also more likely to be found in certain conditions. Pelagic stage juveniles were significantly more likely to be kept as pets than other life stages (OR = 124.56; 95%CI: 34.33–862.49; p = <0.0001) and juvenile animals were more likely to be found entangled than adults (OR = 1.92; 95%CI:1.07–3.58; p = 0.028). No significant relationship was found between life stage and odds of being found beached, floating, or for animals with an unknown status (Fig 3C).

## Causes of morbidity and mortality

Of the injured turtles (n = 379), anthropogenic factors were considered the primary cause of morbidity or mortality in 75.2% of cases (n = 285). The most common source of injury was entanglement in ghost fishing nets or similar materials (66.7%, n = 253) with wounds characteristic of extended periods of net or line entrapment identified in both entangled turtles (n = 215), and in those found floating or beached (n = 38), indicating previous entanglement. Wounds associated with entanglement commonly included lacerations to one or more of the flippers (63.6%, n = 124 of 195 turtles where flipper injuries were characterised), as well as

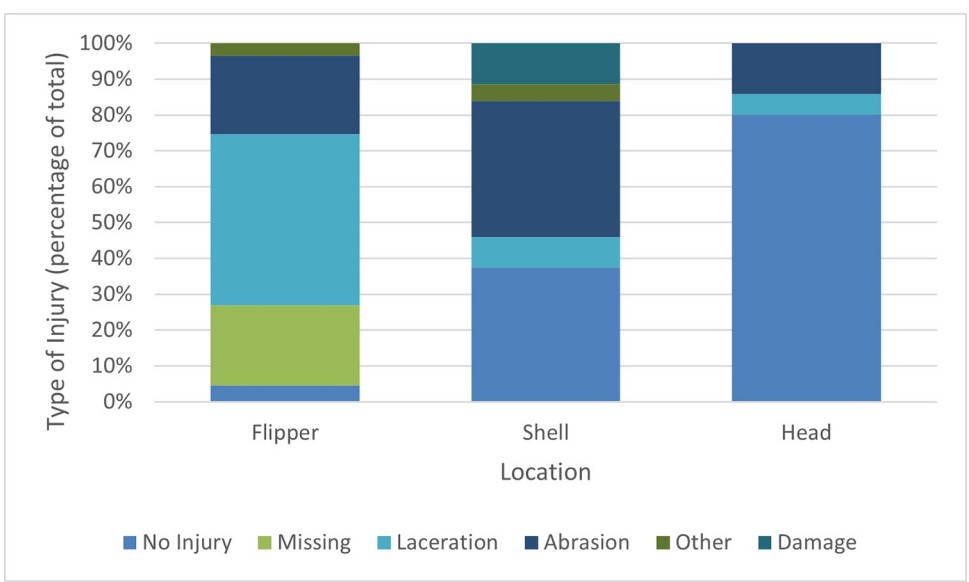

**Fig 4. Summary of injuries caused by entanglement.** NB. Injuries to flippers were recorded based on the presence of a particular injury type on one or more flippers. If multiple injury types were present on different flippers both were recorded.

traumatic amputations (29.7%, n = 58), abrasions (29.2%, n = 57), and other injuries (4.6%, n = 9). Damage to the carapace or plastron was also common (59.4%, n = 116/195 records) and consisted of surface abrasions (41.0%, n = 80) or more severe damage (12.3%, n = 24) including missing scutes or shell fractures. Head injuries were less common (19.9%, n = 38/191 records) and were predominantly abrasions (14.1%, n = 27) with some deeper lacerations (5.7%, n = 11). Abnormal positive buoyancy was also recorded in 52.6% of entangled turtles (n = 93/171 records) (Fig 4).

Turtles found floating or beached were mostly stranded for unknown reasons (47.7% n = 64/134 cases). Excluding those stranded as a result of previous entanglement (29.8%, n = 40) causes of stranding were varied and included cachexia (n = 8), infection (n = 6), gastro-intestinal obstruction (n = 1), congenital abnormality (n = 1), and tar ingestion (n = 1).

Of the 26 turtles kept as pets, 67.8% had health issues associated with poor husbandry (n = 19) including bites from conspecifics (n = 15), infections (n = 13), nutritional deficiencies (n = 3), shell deformities (n = 2), gastro-intestinal blockages (n = 1), and limb fractures (n = 1).

In total, 9 turtles showed evidence of boat strike. In 5 cases this was considered the primary cause of stranding with the remainder occurring concurrently with entanglement. Injuries associated with boat strikes were characterised as often severe damage or lacerations to the carapace (100%, n = 7/7 cases where injuries were characterised), and wounds to the flippers, (71%, n = 5) such as abrasions (28%, n = 2) and traumatic amputations (42%, n = 3).

Additionally, 4 turtles were found with hook injuries (0.87% of total cases) and 2 were found dead after being poached (0.43%).

## Outcomes of injuries

Overall, 65.1% of turtles found alive were released (n = 259/398), 6.3% were transferred to other facilities (n = 25) and 4.6% did not have their outcomes recorded (n = 18). Of the 303 rehabilitation cases with a recorded outcome there were 100 mortalities, giving an overall mortality rate during rehabilitation of 33%.

Of the 57 turtles which were found dead, 47.4% were mortalities of unknown cause (n = 27), 38.6% were associated with entanglement, and the remainder had varied causes including poaching (3.5%, n = 2), boat strike (3.5%, n = 2), cachexia (3.5%, n = 2), and blunt trauma (1.7%, n = 1).

Entanglement carried a better prognosis (OR = 3.58; 95%CI;2.16–5.95; p = <0.0001, 19.9% mortality rate, n = 57/286) than for other causes of injury. There was no significant difference in mortality rates between other groups ($X^2$ = 0.136, df = 3, p = 0.99) (Turtles found floating 48.1% mortality, n = 25/52, beached 50.0%, n = 3/6, kept as pets 44.4%, n = 12/27, boat strike 50%, n = 3/6).

Turtles found alive reached an outcome on average 70 days after being found (0–1434 days, SD:153.1). Time taken to reach an outcome was not significantly different between turtles which were re-released and those which died (t = 0.79, df = 375, p = 0.425). However, time taken to reach an outcome varied between causes of admission ($X^2$ = 704.65, df = 8, p = <0.0001) with cases of entanglement (mean time in rehabilitation = 68 days), and animals kept as pets (mean = 52 days) reaching an outcome more quickly than floating turtles (mean = 111 days), boat strikes (mean = 120 days), and beached turtles (mean = 239 days).

## Seasonality

Frequency of stranded and injured turtles varied between seasons ($X^2$ = 82.88, df = 3, p = <0.0001) with more reports occurring during the northeast monsoon (Jan-Mar, n = 196) than in other seasons (Apr-June inter-monsoon period, n = 100, southwest monsoon (Jul-Sept), n = 97, Oct-Dec inter-monsoon period n = 66) (Fig 5). However, whilst frequency of reports varied between seasons for olive ridley turtles ($X^2$ = 129.62, df = 3, p = <0.00001) with 51.0% of admissions occurring during the northwest monsoon, rates of hawksbill morbidity and strandings remained constant throughout the year ($X^2$ = 0.971, df = 3, p = 0.808). Although reports of green turtles appeared to vary throughout the year ($X^2$ = 14, df = 3, p = 0.0029), this was biased by 14 turtles which were confiscated on the same day after having been kept as pets and which originated from a single source. When these were counted as a single event there was no variation in frequency between seasons for this species ($X^2$ = 1.75, df = 3, p = 0.626).

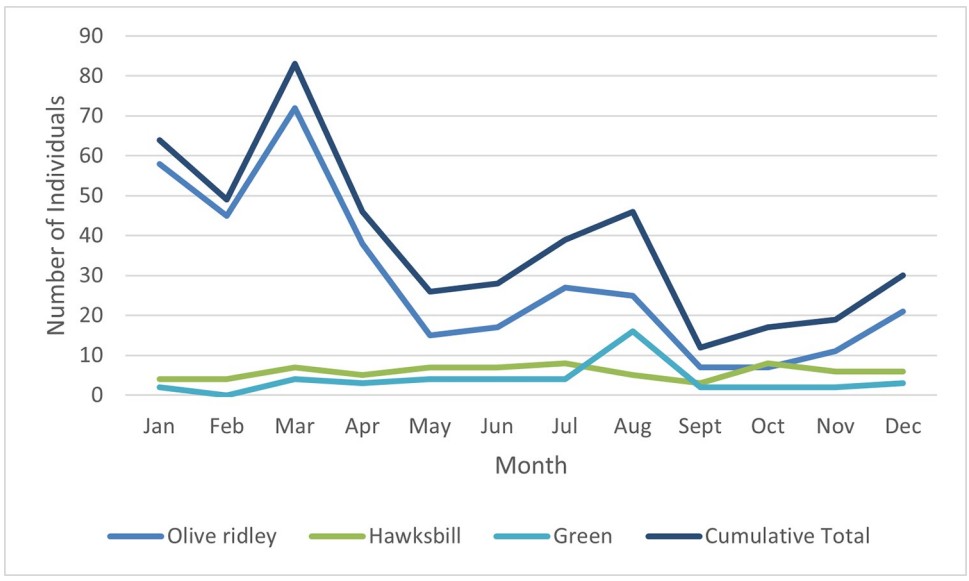

**Fig 5. Stranding and rehabilitation cases by month reported and species.**

## Discussion

By evaluating causes of morbidity and mortality in stranded sea turtles, we present important information regarding threats to local populations. The use of data gathered from the post-stranding rehabilitation process in this study supplements long-term stranding records to provide a more complete assessment of anthropogenic threats and facilitates the first comprehensive evaluation of sea turtle morbidity and mortality in this region of the Indian Ocean over a protracted time period. The inclusion of injury outcomes, along with causes of morbidity and mortality not covered by stranding data, i.e., turtles which had been kept as pets, provides a more complete picture of the nature and magnitude of threats to local sea turtle populations, and will facilitate more informed conservation planning within the region.

Here, we show the overall demographics of stranded turtles concur with findings from other global regions. Juvenile animals were recorded more frequently than adults and, within the group of adult animals whose sex was determined, females were more abundant than males. These trends are common to strandings and in-water observations of marine turtles in multiple regions, including the Maldives, and represent overall population demographics [27,31,45]. Although a predominance of juveniles within turtle populations is a common finding, our study recorded no adult hawksbill turtles. Size of adult turtles is known to vary between regions, and it has been suggested that both green and hawksbill adults may be smaller in the Maldives than in surrounding regions [23]. However, as no definitive measurements of nesting adults are available for this region, utilising measurements taken in other regions, as was done here, may result in an underestimation of adults. Comparatively, proportions of adult green turtles found here correspond to a large dataset collected from Mexico, suggesting that the adult: juvenile ratio for this species may be representative [13]. Further work is required to confirm these observations.

Despite not residing within the Maldives, olive ridley turtles were found stranded at a significantly higher frequency than green and hawksbill turtles. The most abundant of all marine turtle species, olive ridley are known to have large breeding populations around India and Sri Lanka [29,44,49]. It has previously been suggested that the high numbers of stranded olive ridley turtles seen in the Maldives are the result of a large offshore population coinciding with the strong monsoon currents which sweep through the atoll chain, carrying any debilitated animals caught in them into the atolls [31,32].

Globally, fisheries are considered the greatest threat to sea turtle populations [32]. Bycatch is recorded as a key cause of morbidity and mortality of marine turtles in many regions with interactions with trawl, longline, or gillnet operations considered to have the highest overall impacts [50]. However, as these practices are not permitted within the study area [51], here reports of bycatch are very low. Instead, we find entanglement in discarded fishing gear to be the most frequent cause of turtle injuries and mortalities overall (66.2% of all cases). Comparatively, most other studies investigating causes of sea turtle morbidity report lower rates of entanglement [52–55].

Entanglement rates here are also higher for olive ridley turtles than other species. Predisposition of olive ridley turtles in the Indian Ocean to entanglement has previously been attributed to mass nesting aggregations (arribadas) which occur on an annual basis along the east coast of India between December and March, coinciding with peaks in fishing activity in the same area to create "entanglement hotspots" [32,49]. As a predominantly pelagic-living species, olive ridley turtles also encounter accumulations of marine debris which occur within the ocean fronts used to migrate and forage, increasing their risk of entanglement [31,32,49]. Green and hawksbill turtles are found entangled less frequently most likely as they spend the majority of their time on neritic reefs where densities of marine debris are lower, only

transiting through pelagic fronts as young juveniles and as adults during breeding migrations [25,56].

Although the very low rates of bycatch and injuries caused by active fishing found here are inconsistent with global findings, these do correspond with national records [51]. In the Maldives, strict regulations limit commercial fisheries almost exclusively to bait, handline, and pole-and-line operations targeting predominantly tuna species. These techniques are associated with low levels of bycatch in comparison with other fishing methods [51,57]. In many other global regions, fishing techniques associated with high levels of bycatch, such as trawl, gillnet, and longline fishing, predominate [58]. Although reports of turtle bycatch are rare in the Maldives, this is not the case in other species. A recent study of sublethal injuries in Maldivian reef manta rays *(Mobula alfredi)*, which examined over 73,000 photo-identification images, found hook and line injuries to be the leading cause of morbidity in this species [59]. The large body size and high mobility of reef manta rays may increase the likelihood of encountering fishing lines compared with turtles. However, as poor compliance has previously been noted in Maldivian fisheries, resulting in the repeated suspension of longline fishing [51], it is also possible that turtle bycatch is under-reported and, therefore, under-represented here.

The second most frequent stranding presentation, after entanglement, was of floating and beached turtles. Whilst a cause of morbidity is attributed to some of these cases, such as previous entanglement or boat strikes, cause of stranding for the majority remains undetermined. Causes of marine turtle strandings are often difficult to discern due to rapid autolysis of dead animals, limited access to resources such as diagnostic medical equipment, and financial or time constraints. As a result, sea turtle stranding studies frequently report a high proportion of cases with an unknown cause of morbidity or mortality [12,60]. Here, however, this proportion is small with cause of stranding remaining unknown in only 12.8% of cases. Comparative studies report between 20 and 86% of strandings as having an undetermined cause (average between studies48.6%) [12,22,60–63]. Here, the high proportion of cases with distinctive entanglement injuries leaves smaller numbers with their root cause undetermined.

Turtles which had been kept as pets were the third most frequent presentation. Although freshwater turtles are commonly kept as pets around the world [64], records of sea turtles kept domestically are scarce [65]. In the Maldives however, the practice of collecting turtle hatchlings from nesting beaches, raising them for several months and subsequently releasing them has historically been commonplace (*pers comm*, *EPA)*. Despite both green and hawksbill turtles nesting in the Maldives only green turtles were recorded as being kept as pets. This is potentially explained by green turtles being less adverse to nesting in and around areas of human disturbance, making their nests more easily locatable [27]. As protected species, keeping turtles in a captive environment was banned under the Environmental Protection and Preservation Act of the Maldives (2016) with the exception of registered rehabilitation centres. Despite this, the practice continues to an extent (*pers comm*, *EPA)*. Turtle hatchlings held in inappropriate environmental conditions can develop a variety of health complications, resulting in their admittance to rehabilitation.

Boat strikes are a commonly noted cause of anthropogenic injuries in marine turtles, particularly in areas with high levels of water-based traffic [8,12,66]. Here we find boat strikes account for 2.4% of injuries overall, a figure comparable to other regions [8,12]. The not-insignificant risk posed by boat strikes to sea turtles has led to restrictions of water-traffic movements in areas of high turtle activity in several countries [67,68]. Although similar policies are currently implemented in the Maldives for other megafauna species, eg. manta rays and whale sharks [69], currently no such delineations exist for turtles. Further work is required in this regard to identify areas with high turtle activity and establish the risk presented by boat strikes within these zones.

Records also note several cases of poaching. Targeted exploitation of marine turtles to produce commercial products or for consumption is a contributing factor in the decline of all assessed marine turtle species [9]. In Mexico's Bahia Magdalena region, where sea turtle meat is considered a delicacy, intensive over-exploitation throughout the mid 1900's led to a dramatic reduction in turtle populations. Despite a complete ban on turtle catch, use and trade in 1990, in 2006 it was estimated targeted exploitation still accounted for between 63 and 91% of total mortalities. Similarly, in the Maldives turtle meat and products form part of the traditional culture and diet. Although exploitation historically occurred on a much smaller scale than was seen in Mexico, the practice was still considered to negatively affect population numbers and was banned in 2016. However, a 2020 survey conducted by researchers from the Environmental Protection Agency (EPA) found that turtle meat and egg consumption is still prevalent, particularly among younger generations [70]. It is likely that the very low numbers of poached carcasses recorded here are an under-representation due to discrete disposal of carcasses limiting numbers of confirmed cases and low observation effort due to difficulties policing such an extensive and sparsely populated region.

Despite the common occurrence of conditions such as fibropapillomatosis (FP) and spirorchiid infection in global sea turtle populations, no evidence of either condition was recorded here [12,71–73]. FP is an emergent herpesvirus first documented in Florida in the 1980's [74] which affects all species of marine turtle and has spread over time to multiple regions including the Americas, Australia, Indonesia, and East Africa [6,71,73]. The virus can affect a large proportion of a population and is implicated as both a contributing and causal factor of sea turtle strandings, morbidity, and mortality through the production of internal and external neoplasms [71]. Although spread is thought to occur through direct contact or fomites and the disease is present in neighbouring populations [16], as yet no cases have been reported within the Maldives. As the virus produces highly characteristic and often conspicuous pathology it is likely that the condition is not currently present within this region. However, as future introduction is possible, surveillance for the virus is recommended.

Similarly, no cases of spirorchiid infection are observed here. Spirorchiid trematode infection is a bloodborne parasite with high prevalence in sea turtle populations in multiple global regions [[14,75]. Pathology is primarily associated with vascular lesions including aneurysm, arteritis, endocarditis, haemorrhage, thrombosis, and granulomatous inflammation [75]. However, the true significance of spirorchiidosis in sea turtles remains unclear; whilst some sources cite infection as a major cause of debilitation and stranding [14], others surmise burdens are largely incidental [76]. Although spirorchiid infection has never been reported in the Maldives and no evidence was found of it in this study, clinical signs are generalised and pathology subtle. As veterinary pathology is in its infancy in the region it is possible that spirorchiidosis is present but remains undetected. As with FP, active monitoring for this condition is recommended.

Finally, no cases of cold shock are recorded here. Cold shock is a condition physiologically similar to hypothermia in which turtles become immobilised and stranded when exposed to low water temperatures [77]. Most frequently reported in the more temperate waters of Europe and North America, the tropical climate of the Maldives does not reach cold enough temperatures for cold shock to occur [77,78].

In addition to investigating causes of morbidity and mortality, we also identified a seasonal pattern in strandings with an overall peak during the northeast monsoon (Fig 4). Variation in the frequency of sea turtle strandings can be caused by various mechanisms including natural seasonal variation in turtle distribution, environmental events such as storms, or peaks in anthropogenic threats such increased risk of bycatch during fishing seasons or boat strikes during tourist high seasons [13,22,52,60,77]. An increase in the frequency of olive ridley

strandings has previously been recorded within the Maldives and attributed to the peak of the northeast monsoon currents coinciding with arribada nesting on the east coast of India [31].

One further notable point concerns the identity of the people reporting stranded or injured turtles. Overall, 93% of turtles were reported by parties directly involved in tourism, with only 4% originating from local stakeholders (e.g. fishermen). This large disparity may be a result of differences in environmental awareness, including of conservation issues and options for reporting, between the 2 groups [79]. Many tourist operations in the Maldives are environmentally aware, promote sustainability, and are aligned with conservation objectives [80]. Conversely, other sectors such as fisheries are more economically focused and may therefore be less likely to report stranded or injured animals [34].

As the first comprehensive, long term, and multispecies analysis of morbidity and mortality in the region, this study has identified several points of critical importance to marine turtle conservation in the Indian ocean. In contrast to other stranding studies, here we identify ghost nets as the leading cause of injuries and mortalities, implicated in 66.2% of all reported cases. Abandoned, lost, or discarded fishing gear contributes significantly to marine plastic pollution, estimated to make up around 10% of all marine litter, and is a major global threat to all types of marine megafauna [81,82]. As the Maldives covers a comparatively tiny area of the Indian ocean, and reported turtles account for a small proportion of those affected, the scale of ghost fishing will be far more extensive than reported here [21]. Although it is important to note that entanglement cases are likely to be over-represented in this study as the often-buoyant ghost nets will increase the likelihood of affected individuals being recovered compared with other causes of debilitation, it should also be considered that this bias will equally affect other global regions. The comparative scale of entanglement cases identified here should not be underestimated. Tackling the global issue of ghost fishing requires extensive international legislation and co-operation to clean up current pollution and limit further additions.

In addition to highlighting the impacts of ghost nets, this study has also identified several key gaps in current knowledge. Determining the morphology of adult animals, evaluating the true scale of both bycatch and targeted capture of marine turtles, and developing disease surveillance strategies for both FP and spirorchiidosis have all been identified as areas for future study. Further work in these areas will help to build a more accurate picture of the status of sea turtle populations in this region. Additional work to identify causes of and address low stakeholder engagement could help increase the numbers of injured and stranded turtles which are reported and treated.

To conclude, the Indian Ocean is a region containing important sea turtle habitat but which remains comparatively under-studied. This work has identified the major causes of morbidity and mortality within the Maldives and has determined several avenues for future study which will greatly improve understanding of local populations to assist in their conservation. However, for this work to impact practical conservation, it is critical that relevant findings are applied to procedure and policy both nationally and internationally.

## Supporting information

**S1 Appendix. Dataset.**
(XLSX)

## Acknowledgments

The authors would like to extend their gratitude to Four Seasons Resorts Maldives at Landaa Giraavaru and Kuda Huraa for their continued support of the Maldives Sea Turtle Conservation Program and other research conducted under the Marine Savers initiative.

## Author Contributions

**Conceptualization:** Katrina Himpson.

**Data curation:** Katrina Himpson.

**Formal analysis:** Katrina Himpson.

**Investigation:** Katrina Himpson.

**Methodology:** Katrina Himpson.

**Project administration:** Katrina Himpson.

**Supervision:** Thomas Le Berre.

**Writing – original draft:** Katrina Himpson.

**Writing – review & editing:** Simon Dixon.

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
