## [Decision Letter · Decision Letter 0]

20 Jan 2023

PONE-D-22-34145Evaluation of sea turtle morbidity and mortality within the central Indian Ocean from 12 years of data shows high prevalence of ghost net entanglement.PLOS ONE

Dear Dr. Himpson,

Thank you for submitting your manuscript to PLOS ONE. After careful consideration, we feel that it has merit but does not fully meet PLOS ONE’s publication criteria as it currently stands. Therefore, we invite you to submit a revised version of the manuscript that addresses the points raised during the review process.

We look forward to receiving your revised manuscript.

Kind regards,

Graeme Hays

Academic Editor

PLOS ONE

Journal Requirements:

**Additional Editor Comments:**

We now have three detailed and constructive reviews of your manuscript. You will see that the referees liked aspects of the manuscript but there are several concerns. You’ll see that the referees consistently thought that your writing was a little too parochial and the results poorly presented. On balance I think that if you take care with a major revision, this manuscript might still be suitable for PLoS1 and so I am recommending that you revise the manuscript taking all the comments into consideration. I expect that this major revision will take some time as all the referees called for some substantive changes.

I look forward to seeing a revision.

All best wishes, Graeme Hays

Reviewers' comments:

Reviewer's Responses to Questions

**Comments to the Author**

1. Is the manuscript technically sound, and do the data support the conclusions?

Reviewer #1: No

Reviewer #2: Yes

Reviewer #3: Partly

2. Has the statistical analysis been performed appropriately and rigorously? 

Reviewer #1: No

Reviewer #2: Yes

Reviewer #3: No

3. Have the authors made all data underlying the findings in their manuscript fully available?

Reviewer #1: No

Reviewer #2: No

Reviewer #3: No

4. Is the manuscript presented in an intelligible fashion and written in standard English?

Reviewer #1: No

Reviewer #2: Yes

Reviewer #3: Yes

5. Review Comments to the Author

Reviewer #1: This study presents an interesting insight on the prevalence of ghost-fishing in the Maldives, driving strandings of sea turtles. However, at present, this work is not placed in sufficient context of the wider literature, limiting the interest of the readership. Also, no figures/tables were provided, making interpretation very difficult. I have made detailed recommendations up to the end of the results; however, I have not assessed the discussion, due to the extent of revision required throughout. With careful and detailed revision, this could make a very interesting contribution to the journal.

Abstract

Line 25-7; this is too broad; what is the key knowledge gap that needs to be explored in general? Maybe amend to something like “Quantifying how ghost fishing contributes to anthropogenic-driven losses of sea turtles could help guide marine management; however…” why has this not been done so far, what limitations have stopped this, why do you have the tools to achieve it now?

Line 27-8, this is melodramatic and again too broad; what information is missing in this region, and why, due to a lack of funds, equipment, technology? Clarify.

Line 29 – separate your aim from your key finding

“Here, we utilised 12 years of stranding and rehabilitation data from the Republic of the Maldives to

identify key threats to marine turtles in this region.”

State which 12 years and whether this coverage was nationwide or biased.

Then state the results

“Entanglement in ghost nets represented the leading cause of injuries and strandings (66.2% of cases).”

It is not clear what the “regions” you were referring to here, if you want to make a comparison to the wider literature, this should be done in the discussion, as here it would mean your needing to add citations, which is not permitted. The Abstract should present your results.

Line 33. Make this a new sentence, and avoid referring to the wider literature.

Line 35, what do you mean by “non resident” and “resident”– non resident to where, as you state the study was done at a national scale? This does not have context. The reader must be informed of why you are using these different terms. This should be your first statement of your Abstract results, then percentage of human causes, then the bycatch vs ghost fishing.

Line 37-38, this result while interesting, does not contribute to your narrative here; remove it, or give some interpretation of why it is relevant to the ghost fishing bias.

“and identifies several

40 key areas where current knowledge of sea turtle ecology in this region is lacking.” – delete, this does not convey useful information; it is a throwaway statement that could be placed on any paper.

“However, for the

41 findings of this study to impact the status of marine turtles they must be implemented into practice and

42 policy.” – also delete this, it weakens your final statement of the previous sentence.

Keywords – missing

Introduction

Please check that citations are formatted correctly.

This entire section requires careful revision. The introduction is too long and too turtle centred. There are too many short paragraphs with no clear message. I advise that this section is carefully revised taking the broad readership of the journal into consideration.

I would advise the following

Paragraph 1 – What is the current prevailing threat to marine vertebrates globally, turtles are one of several groups threatened by human activities at sea. There is a vast global literature on this, and you can identify ghost fishing as one such component here and how it compares to other human caused mortality.

Paragraph 2 – Ghost fishing – focus on detail on this, and whether threats are uniform globally across marine vertebrates, and if not identify potential reasons why not.

Paragraph 3 – Sea turtles and threats at sea; draw on the wide literature of human caused threats at sea (see Wallace et al. 2010 and associated publications), and in particular explore the literature around ghost fishing and sea turtles; if it is limited, here, highlight why, what approaches work best with detecting this effect, i.e. draw on rehabilitation and strandings and the pros/cons of using such literature.

Paragraph 4 – your study aims and hypotheses, along with main approach, study region. What did you expect to find?

Line 102 – “by comparing these findings to other regions” – this statement needs careful reconsideration; this implies you conducted an exhaustive literature review; if this is the case, then in your Paragraph 4, where you state your aims, you also need to state that your work is a combination of both empirical data and a wider literature review. This must be clarified in the abstract and the exact approach of the review stated in the methods.

Line 104-5, this is fine

Line 106 – it is not clear how this is feasible, remove

Methods

Line 110, just put the coordinates in parentheses at the end of the last sentence.

A figure is required showing the entire nation and position of the two atolls, along with the catchment area for strandings assessed here.

No tables or figures are provided on the manuscript, making evaluation difficult.

Line 128 – what are circumstantial data, give some examples of what this includes.

Line 144 – at the site or on arrival? If at the site, how was this standardized across different people finding animals?

Change disposition to status throughout.

Line 162, were necroposies done to check for blockages? If not, you cannot state if this is natural.

Line 163, it is not clear why husbandry fits here. At stranding, husbandry is not relevant; if they survive rehabilitation cannot really be determined as a husbandry issue. The context of husbandry needs careful consideration. How do you assess if husbandry was poor?

You do not state anywhere what area your strandings covered – is it national? How do you overcome bias to populated areas?

Results

Supporting tables and figures are required.

This needs to be broken into 3-4 key subsections each supported by a clear figure. Shift all percentages into tables, then revise your entire text to focus on the key points of interest.

By delineating clear subsections, you can then develop some clear hypothesis testing.

Line 185, how do you get a female bias for adult hawksbills if no adults were recorded?

Line 185, you simply state, The adult sex ratio was X:X for green turtles and X:X for olives. What is this sex ratio based on, the stranded animals? If something else, more explanation of how it was obtained is required in the methods.

Line 188-191, what about inhabited vs uninhabited islands?

A map showing the distribution of all strandigns for the 3 species (different colours) and size classes should be presented, along with highlighting inhabited islands.

Line 192, while turtles were found entangled in nets, this might easily be just the end point, i.e. turtle gets infection, floats, becomes entangled; entanglement itself might not be the primary cause.

How do you address this? For instance, all animals could have been floating, then become entangled; the key point is to find the initial cause for floating.

Line 192-5, this needs to be presented graphically.

Graphs are required to support the data throughout this section; once you have graphs, you can then interpret the key points of interest, rather than listing everything.

Line 219 – were necropsies/xrays conducted?

Reviewer #2: Here the causes of injury and reasons for mortality are reported for turtles found stranded or floating in the Maldives, Indian Ocean. Is it concluded that entanglement in ghost nets is a key source of injury and mortality.

This is an interesting manuscript that I enjoyed reading. Some nice data are presented. I have some suggestions to improve the final version. With a little care, these revisions should be straightforward to deal with.

1. Line 25. “Anthropogenic activities are driving the global decline of sea turtle populations.

Sensationalist as well as incorrect. You need to be a little more honest.

See:

Mazaris AD et al. (2017). Global sea turtle conservation successes. Science Advances 3: e1600730. https://doi.org/10.1126/sciadv.1600730

I think throughout (e.g. start of Intro, lines 45-52) you need to say that at many sites around the world, sea turtle numbers are increasing due to conservation efforts (Mazaris). But regardless, sea turtles still face threats and so to facilitate further population increases it is important to both identify and then mitigate threats.

2. Line 34. “ … causes of admission …”

Reads poorly as, in the abstract, you have not mentioned anything about admission to a rehab centre being the basis for the data collection.

3. Line 77. “Despite having large populations of sea turtles, the central Indian Ocean is one region where evaluation of threats has been neglected …”

Can you add some details about “large populations”. E.g. see work of Jeanne Mortimer and others for regional assessments of nesting numbers – see: https://doi.org/10.1017/S0030605319001108

4. Line 85. “However, numbers of both species are currently declining.”

Can you qualify this statement ? Where is this info published ?

5. Line 138. “Green turtles with CCL greater than 95cm (males) and 100cm (females) were classed as adults …”

Can you estimate the probability of a mis-id of life stage, e.g. a sub-adult male being classified as a female ? This is important as later you report sex ratios, but I suspect you’ll have quite a few mis-ids.

6. Lines 162. “… cachexia (emaciation with no discernible primary cause …”

How was this recorded ? e.g. some threshold residual value from a length:weight relationship ? Or some objective scoring (e.g. see Heithaus/ Thomson work in Shark Bay) or did you just make a subjective best guess ? That’s fine. Just explain what you have done.

7. Line 207. “… was entanglement in ghost fishing nets or similar materials (66.2%, n=251) with wounds characteristic of net or line entrapment identified in both entangled turtles (n=211) and in those found floating or beached (n=40) …”

If a turtle just has wounds characteristic of nets or lines, how do you know this was a “ghost net” and not just a net being used by a fisherman who released the turtle ?

8. Line 226. Boat strike. Was this associated with nearby resorts ?

I would have a few lines of the Discussion about boat strikes. This has been widely reported around the world (e.g. see Gail Schofield’s work in Greece, https://doi.org/10.1002/ecy.3027 and also in the USA, see Tony Tucker). Boat strike is particularly associated with high boating traffic and had led to speed restrictions in high use turtle areas in the US and Greece.

9. Line 286. You mention where Olive Ridley likely may have come from. Similarly can you detail some of the areas that adult greens and hawksbills have come from ? There has been satellite tracking showing adult green from the Chagos Archipelago travel to the Maldives. But adult hawksbills from Chagos likely do not travel to the Maldives … rather all the tracked animals stay within Chagos (e.g. see https://doi.org/10.1098/rsif.2021.0859). So I guess any adult hawksbills you see also nest in the Maldives ?

10 There are no figures. Perhaps think of 1 or 2 key results to show in a graph ?

e.g. a map showing the number of reports came from different places in the Maldives. Perhaps a photo of a stranded turtle in a ghost net.

11. “Poaching”. I could not see anything in the methods about how this was ascertained. Describe your method. Include a photo ?

In summary, a nice piece of work with some interesting results. Well done to the authors on completing such a nice study.

Graeme Hays

Reviewer #3: See attached review with suggestions to improve the manuscript.

It was difficult to state that the analysis was appropriate when no supporting data were provided and no tables or figures were included.

6. PLOS authors have the option to publish the peer review history of their article (what does this mean?). If published, this will include your full peer review and any attached files.

Reviewer #1: No

Reviewer #2: No

Reviewer #3: No

---

## [Author Response · Author response to Decision Letter 0]

21 Jun 2023

Formatting edited to meet requirements.

Minimum dataset included with this draft

Minimum dataset included with this draft

Additional Editor Comments:

We now have three detailed and constructive reviews of your manuscript. You will see that the referees liked aspects of the manuscript but there are several concerns. You’ll see that the referees consistently thought that your writing was a little too parochial and the results poorly presented. On balance I think that if you take care with a major revision, this manuscript might still be suitable for PLoS1 and so I am recommending that you revise the manuscript taking all the comments into consideration. I expect that this major revision will take some time as all the referees called for some substantive changes.

I look forward to seeing a revision.

Reviewers' comments:

Reviewer's Responses to Questions

Comments to the Author

1. Is the manuscript technically sound, and do the data support the conclusions?

Reviewer #1: No

Reviewer #2: Yes

Reviewer #3: Partly

Data collection was performed as rigorously as possible over the time period

2. Has the statistical analysis been performed appropriately and rigorously?

Reviewer #1: No

Reviewer #2: Yes

Reviewer #3: No

Statistical analysis kept relatively simple and based on observations only

3. Have the authors made all data underlying the findings in their manuscript fully available?

Reviewer #1: No

Reviewer #2: No

Reviewer #3: No

Data included in this revision

4. Is the manuscript presented in an intelligible fashion and written in standard English?

Reviewer #1: No

Reviewer #2: Yes

Reviewer #3: Yes

Manuscript edited based on suggested grammatical revisions. Proofread for further grammar and typographical errors.

5. Review Comments to the Author

Reviewer #1: This study presents an interesting insight on the prevalence of ghost-fishing in the Maldives, driving strandings of sea turtles. However, at present, this work is not placed in sufficient context of the wider literature, limiting the interest of the readership. 

Introduction refocused to broaden interest. Discussion is already within a global context

Also, no figures/tables were provided, making interpretation very difficult.

Figures included to clarify results

I have made detailed recommendations up to the end of the results; however, I have not assessed the discussion, due to the extent of revision required throughout. With careful and detailed revision, this could make a very interesting contribution to the journal.

Abstract

Line 25-7; this is too broad; what is the key knowledge gap that needs to be explored in general? Maybe amend to something like “Quantifying how ghost fishing contributes to anthropogenic-driven losses of sea turtles could help guide marine management; however…” why has this not been done so far, what limitations have stopped this, why do you have the tools to achieve it now?

Reworded and clarified

Line 27-8, this is melodramatic and again too broad; what information is missing in this region, and why, due to a lack of funds, equipment, technology? Clarify.

Clarified

Line 29 – separate your aim from your key finding

“Here, we utilised 12 years of stranding and rehabilitation data from the Republic of the Maldives to

identify key threats to marine turtles in this region.”

Reformatted

State which 12 years and whether this coverage was nationwide or biased.

Then state the results

“Entanglement in ghost nets represented the leading cause of injuries and strandings (66.2% of cases).”

It is not clear what the “regions” you were referring to here, if you want to make a comparison to the wider literature, this should be done in the discussion, as here it would mean your needing to add citations, which is not permitted. The Abstract should present your results.

Reworded

Line 33. Make this a new sentence, and avoid referring to the wider literature.

Done

Line 35, what do you mean by “non resident” and “resident”– non resident to where, as you state the study was done at a national scale? This does not have context. The reader must be informed of why you are using these different terms. This should be your first statement of your Abstract results, then percentage of human causes, then the bycatch vs ghost fishing.

Clarified

Line 37-38, this result while interesting, does not contribute to your narrative here; remove it, or give some interpretation of why it is relevant to the ghost fishing bias.

Removed

“and identifies several

40 key areas where current knowledge of sea turtle ecology in this region is lacking.” – delete, this does not convey useful information; it is a throwaway statement that could be placed on any paper.

Removed

“However, for the

41 findings of this study to impact the status of marine turtles they must be implemented into practice and

42 policy.” – also delete this, it weakens your final statement of the previous sentence.

Removed

Abstract reworded taking all comments into consideration

Keywords – missing

Keywords added

Introduction

Please check that citations are formatted correctly.

One citation reformatted

This entire section requires careful revision. The introduction is too long and too turtle centred. There are too many short paragraphs with no clear message. I advise that this section is carefully revised taking the broad readership of the journal into consideration.

Introduction rewritten based on below comments and suggestions

I would advise the following

Paragraph 1 – What is the current prevailing threat to marine vertebrates globally, turtles are one of several groups threatened by human activities at sea. There is a vast global literature on this, and you can identify ghost fishing as one such component here and how it compares to other human caused mortality.

Paragraph 2 – Ghost fishing – focus on detail on this, and whether threats are uniform globally across marine vertebrates, and if not identify potential reasons why not.

Paragraph 3 – Sea turtles and threats at sea; draw on the wide literature of human caused threats at sea (see Wallace et al. 2010 and associated publications), and in particular explore the literature around ghost fishing and sea turtles; if it is limited, here, highlight why, what approaches work best with detecting this effect, i.e. draw on rehabilitation and strandings and the pros/cons of using such literature.

Paragraph 4 – your study aims and hypotheses, along with main approach, study region. What did you expect to find?

Introduction rewritten based on the above suggestion to broaden the context

Line 102 – “by comparing these findings to other regions” – this statement needs careful reconsideration; this implies you conducted an exhaustive literature review; if this is the case, then in your Paragraph 4, where you state your aims, you also need to state that your work is a combination of both empirical data and a wider literature review. This must be clarified in the abstract and the exact approach of the review stated in the methods.

Clarified in text and reworded

Line 104-5, this is fine

Line 106 – it is not clear how this is feasible, remove

Removed

Methods

Line 110, just put the coordinates in parentheses at the end of the last sentence.

Revised

A figure is required showing the entire nation and position of the two atolls, along with the catchment area for strandings assessed here.

No tables or figures are provided on the manuscript, making evaluation difficult.

Figures included (map of the location of the Maldives including rehab centres and stranding sites, summaries of admissions and results).

Line 128 – what are circumstantial data, give some examples of what this includes.

Clarified in text. E.g. Date, location

Line 144 – at the site or on arrival? If at the site, how was this standardized across different people finding animals?

Measured on arrival to rehabilitation or on site wherever possible. CCL was left as NA if not measured. Status was recorded on arrival. If the animal was released immediately status was based on the discoverers' description. Clarified in text.

Change disposition to status throughout.

Change made

Line 162, were necroposies done to check for blockages? If not, you cannot state if this is natural.

Necropsies performed wherever possible. However, cases where this was not possible had cause of death recorded as ‘unknown’

Line 163, it is not clear why husbandry fits here. At stranding, husbandry is not relevant; if they survive rehabilitation cannot really be determined as a husbandry issue. The context of husbandry needs careful consideration. How do you assess if husbandry was poor?

Clarified in text. This point refers to cases of turtles being kept as pets as injuries or illness associated with poor husbandry resulted in admission to rehabilitation.

You do not state anywhere what area your strandings covered – is it national? How do you overcome bias to populated areas?

Clarified in text. Stranding reports are national but show bias towards the central, more densely populated atolls.

Results

Supporting tables and figures are required.

Included

This needs to be broken into 3-4 key subsections each supported by a clear figure. Shift all percentages into tables, then revise your entire text to focus on the key points of interest.

By delineating clear subsections, you can then develop some clear hypothesis testing.

Methods broken into subsections. Figures/tables included but numbers left in-text to support written results.

Line 185, how do you get a female bias for adult hawksbills if no adults were recorded?

Error remedied

Line 185, you simply state, The adult sex ratio was X:X for green turtles and X:X for olives. What is this sex ratio based on, the stranded animals? If something else, more explanation of how it was obtained is required in the methods.

Clarified in text. Ratio refers to all stranded individuals for which sex was recorded

Line 188-191, what about inhabited vs uninhabited islands?

See figures for distribution of strandings across atolls. Highlighting inhabited/uninhabited islands not practical as land area is small comparative to total area

A map showing the distribution of all strandigns for the 3 species (different colours) and size classes should be presented, along with highlighting inhabited islands.

Map of stranding sites included in figures. Including sex of the individual made the figure too complicated. Highlighting inhabited/uninhabited islands not practical as islands are too small to be visible on a national, or even an atoll scale.

Line 192, while turtles were found entangled in nets, this might easily be just the end point, i.e. turtle gets infection, floats, becomes entangled; entanglement itself might not be the primary cause.

How do you address this? For instance, all animals could have been floating, then become entangled; the key point is to find the initial cause for floating.

Here we are reporting the initial condition in which animals were found. It is felt that discussing the various causes of buoyancy syndromes in marine turtles is out-with the scope of this paper. However, cataloguing this and other injuries in more detail is currently an area of ongoing work within this dataset.

Line 192-5, this needs to be presented graphically.

Graphs are required to support the data throughout this section; once you have graphs, you can then interpret the key points of interest, rather than listing everything.

Graph of results included. However, as results are broad, including figures of all results would result in a high number of figures being included

Line 219 – were necropsies/xrays conducted?

Necropsies were carried out where possible to determine cause of death. Cases were determined to have an ‘unknown’ cause where necropsies were unable to be performed or where the necropsy results were unconclusive. This has been clarified in the methods section. X-rays were performed in very rare cases only as this involved transporting patients to a human hospital. Until very recently no veterinary x-ray units were available in Maldives. 

Reviewer #2: Here the causes of injury and reasons for mortality are reported for turtles found stranded or floating in the Maldives, Indian Ocean. Is it concluded that entanglement in ghost nets is a key source of injury and mortality.

This is an interesting manuscript that I enjoyed reading. Some nice data are presented. I have some suggestions to improve the final version. With a little care, these revisions should be straightforward to deal with.

1. Line 25. “Anthropogenic activities are driving the global decline of sea turtle populations.

Sensationalist as well as incorrect. You need to be a little more honest.

See:

Mazaris AD et al. (2017). Global sea turtle conservation successes. Science Advances 3: e1600730. https://doi.org/10.1126/sciadv.1600730

Revised and rephrased

I think throughout (e.g. start of Intro, lines 45-52) you need to say that at many sites around the world, sea turtle numbers are increasing due to conservation efforts (Mazaris). But regardless, sea turtles still face threats and so to facilitate further population increases it is important to both identify and then mitigate threats.

Updated in text

2. Line 34. “ … causes of admission …”

Reads poorly as, in the abstract, you have not mentioned anything about admission to a rehab centre being the basis for the data collection.

Reworded to clarify this

3. Line 77. “Despite having large populations of sea turtles, the central Indian Ocean is one region where evaluation of threats has been neglected …”

Can you add some details about “large populations”. E.g. see work of Jeanne Mortimer and others for regional assessments of nesting numbers – see: https://doi.org/10.1017/S0030605319001108

Clarified in text. However, there is currently no data published on the size of turtle populations for this region

4. Line 85. “However, numbers of both species are currently declining.”

Can you qualify this statement ? Where is this info published ?

Since the first submission of this paper Stelfox et al have published evidence that marine turtle populations are stable. This has been updated in the text.

5. Line 138. “Green turtles with CCL greater than 95cm (males) and 100cm (females) were classed as adults …”

Can you estimate the probability of a mis-id of life stage, e.g. a sub-adult male being classified as a female ? This is important as later you report sex ratios, but I suspect you’ll have quite a few mis-ids.

This is addressed in the discussion. Anecdotal evidence suggests that hawksbill and green turtles in this region are smaller than neighbouring populations. However, there is currently no data to back this up.

6. Lines 162. “… cachexia (emaciation with no discernible primary cause …”

How was this recorded ? e.g. some threshold residual value from a length:weight relationship ? Or some objective scoring (e.g. see Heithaus/ Thomson work in Shark Bay) or did you just make a subjective best guess ? That’s fine. Just explain what you have done.

Body condition index calculated using method published by Norton and Wyneken (2015)- clarified in text

7. Line 207. “… was entanglement in ghost fishing nets or similar materials (66.2%, n=251) with wounds characteristic of net or line entrapment identified in both entangled turtles (n=211) and in those found floating or beached (n=40) …”

If a turtle just has wounds characteristic of nets or lines, how do you know this was a “ghost net” and not just a net being used by a fisherman who released the turtle ?

Injuries origin was determined using previously defined criteria (eg. Archibald et al. 2018). The characteristic lacerations caused by extended entanglement would typically not occur with a turtle being line caught; hook injuries to the mouth being more common in these cases. Additionally, in Maldives, methods of fishing which could conceivably cause such injuries such as net or longline fishing, are not permitted (see introduction).

8. Line 226. Boat strike. Was this associated with nearby resorts ?

Very interesting point. All 9 cases were reported by resorts but as resort and local islands exist in close proximity and travelling/drifting would have occurred it would not be possible to determine whether resort or local traffic was the cause of these cases.

I would have a few lines of the Discussion about boat strikes. This has been widely reported around the world (e.g. see Gail Schofield’s work in Greece, https://doi.org/10.1002/ecy.3027 and also in the USA, see Tony Tucker). Boat strike is particularly associated with high boating traffic and had led to speed restrictions in high use turtle areas in the US and Greece.

Short paragraph on boat strikes added to the discussion

9. Line 286. You mention where Olive Ridley likely may have come from. Similarly can you detail some of the areas that adult greens and hawksbills have come from ? There has been satellite tracking showing adult green from the Chagos Archipelago travel to the Maldives. But adult hawksbills from Chagos likely do not travel to the Maldives … rather all the tracked animals stay within Chagos (e.g. see https://doi.org/10.1098/rsif.2021.0859). So I guess any adult hawksbills you see also nest in the Maldives ?

We suspect that the majority of green and hawksbill cases presented to us are resident animals with a few originating from places like Chagos. However, there is currently no data to support this that I am aware of outside of the Chagos post-nesting tracking data.

10 There are no figures. Perhaps think of 1 or 2 key results to show in a graph ?

e.g. a map showing the number of reports came from different places in the Maldives. Perhaps a photo of a stranded turtle in a ghost net.

Noted and included

11. “Poaching”. I could not see anything in the methods about how this was ascertained. Describe your method. Include a photo ?

Clarified in methods

In summary, a nice piece of work with some interesting results. Well done to the authors on completing such a nice study.

Thank you!

Reviewer #3: See attached review with suggestions to improve the manuscript.

It was difficult to state that the analysis was appropriate when no supporting data were provided and no tables or figures were included

Figures, tables and data now included.

This sea turtle study presents 12 years of stranding and rehabilitation data from the Maldives to report key threats to sea turtles, identifying entanglement in ghost nets as one of the key anthropogenic causes of injuries. The dataset is considerable and includes 459 turtles. Clear study objectives are presented including variation in cause of injury between species and life stages, estimation of mortality rate and identification of seasonality in strandings. Providing that the review comments below are addressed, the study would advance our understanding of some of the underreported anthropogenic impacts from fishing, as already highlighted by a number of studies in the region (e.g., Stelfox et al) cited in the manuscript.

The findings are clearly presented but the manuscript could be much improved by re-arranging content into subsections with relevant subtitles in Methods and Results. 

Methods and results re-arranged. Appropriate subheadings included

No figures and/or tables were included so it was difficult to interpret results. Additionally, no underlying data were provided (that I could access) so it was difficult to comment on the review question ‘Have the authors made all data underlying the findings in their manuscript fully available?’. To this end, a supplementary table providing summary data would be helpful to support the growing body of literature on this topic in addition to providing a link to the Reefscapers website.

Figures and tables added along with data

Some specific comments:

Abstract: 

State number of atolls and relative location where data were collected 

Stated in methods and included in Fig 1

The study is for one small area of the Indian Ocean and I suggest re-phrasing ‘high prevalence of ghost net entanglement in the Indian Ocean’. If a review has shown high prevalence elsewhere in the IO then include the results.

Noted and updated in text

Introduction:

I recommend to commence with a paragraph describing different threats to turtles (and introducing the various anthropogenic threats) before focusing on species and assessment.

L77: define what you are including in the term Central Indian Ocean to help with readers’ understanding as typically this region is split between NWIO and SWIO and NEIO with respect to sea turtle regional management units (RMUs) Wallace et al. 2010. http://www.doi.org/10.1371/journal.pone.0015465

Reworded/rephrased throughout for clarity and to fit with accepted nomenclature

L79: insert the citation as numeric.

Reformatted

L84: provide more context about population connectivity in the Western Indian Ocean, for example a number of green turtles have migrated to Maldives from Chagos Archipelago in the south (Hays et al. 2020 https://doi.org/10.1016/j.cub.2020.05.086) which provides critical nesting habitat for turtles from around the region (see Mortimer et al. 2020 http://www.doi.org/10.1017/S0030605319001108). 

Included in introduction

Results:

I would suggest to the authors to provide a number of graphs to show key results, for example: 

To support the statements about turtles stranded or bycaught turtles in the Maldives : provide a map with indication of numbers of each case category at each location

To support statements about seasonal patterns of admissions : provide a graph comparing number of cases per month of the year

To support statements about entanglement being key cause of injuries or death: graph reporting number of cases for each threat identified

Provide a graphical overview of relative proportion of life stages and species included in the study

Figures created and included to illustrate key results

Discussion:

This is quite lengthy and could be improved by re-ordering to commence the section with key results that are discussed in context of the literature, followed by several sections with appropriate sub-titles to discuss the various findings. 

Noted and trialled. The discussion opens with the key finding of the high prevalence of ghost net entanglement. Sub-titles also trialled but it was felt that this broke the discussion into too many short sections.

There are some statements that should only be included if supported by citations (e.g. L418 turtle populations in the Central IO are some of the most threatened in the world …)

Sections rephrased in text

It is worthwhile considering potential bias in the dataset due to method of entanglement. Relative contributions of ghost nets to entanglement and injury/mortality of turtles is possibly overreported compared to other causes of anthropogenic threat. This is because ghost nets are likely to come ashore as they are usually buoyant and carried in surface waters. Turtles that snag on fishing line caught on the reefs are less likely to be included in the dataset. Consider other anthropogenic factors in your discussion, e.g. those reported by Casale et al. 2010 https://doi.org/10.1002/aqc.1133

Point well made; discussion updated to address this.

L294: As suggested for the introduction, a definition of the area (Central Indian Ocean) and a map of the study site would help support this statement. 

Rephrased throughout

---

## [Decision Letter · Decision Letter 1]

28 Jun 2023

PONE-D-22-34145R1Evaluation of sea turtle morbidity and mortality within the Indian Ocean from 12 years of data shows high prevalence of ghost net entanglement.PLOS ONE

Dear Dr. Himpson,

Thank you for submitting your manuscript to PLOS ONE. After careful consideration, we feel that it has merit but does not fully meet PLOS ONE’s publication criteria as it currently stands. Therefore, we invite you to submit a revised version of the manuscript that addresses the points raised during the review process.

Please make the final suggested corrections to the figures to improve their clarity.==============================

We look forward to receiving your revised manuscript.

Kind regards,

Graeme Hays

Academic Editor

PLOS ONE

Journal Requirements:

Additional Editor Comments:

The authors have done a good job with the revisions. For future manuscripts, I would encourage you not to write vague things in your responses letter like “This has been changed in the Discussion”. It is far easier for referees, if you cut and paste in your revised text in the cover letter, so they can quickly see what you have done, rather than having to hint through the manuscript for the changes.

For the histogram figures, I would perhaps not go with subtle differences in shades of blue. This is not very clear. Also note on the y-axis title, “Proportion” means a number from 0 to 1. “Percentage” means a number from 0-100. Also I would give a more information axis title that just “Proportion”. The normal approach when producing a scientific graph is to have an axis title and, in brackets, the unit. “Percentage” is the unit.

Well done. Graeme Hays

Reviewers' comments:

Reviewer's Responses to Questions

**Comments to the Author**

1. If the authors have adequately addressed your comments raised in a previous round of review and you feel that this manuscript is now acceptable for publication, you may indicate that here to bypass the “Comments to the Author” section, enter your conflict of interest statement in the “Confidential to Editor” section, and submit your "Accept" recommendation.

Reviewer #2: All comments have been addressed

2. Is the manuscript technically sound, and do the data support the conclusions?

Reviewer #2: Yes

3. Has the statistical analysis been performed appropriately and rigorously? 

Reviewer #2: N/A

4. Have the authors made all data underlying the findings in their manuscript fully available?

Reviewer #2: Yes

5. Is the manuscript presented in an intelligible fashion and written in standard English?

Reviewer #2: Yes

6. Review Comments to the Author

Reviewer #2: The authors have done a good job with the revisions. For future manuscripts, I would encourage you not to write vague things in your responses letter like “This has been changed in the Discussion”. It is far easier for referees, if you cut and paste in your revised text in the cover letter, so they can quickly see what you have done, rather than having to hint through the manuscript for the changes.

For the histogram figures, I would perhaps not go with subtle differences in shades of blue. This is not very clear. Also note on the y-axis title, “Proportion” means a number from 0 to 1. “Percentage” means a number from 0-100. Also I would give a more information axis title that just “Proportion”. The normal approach when producing a scientific graph is to have a an axis title and, in brackets, the unit. “Percentage” is the unit.

Well done. Graeme Hays

7. PLOS authors have the option to publish the peer review history of their article (what does this mean?). If published, this will include your full peer review and any attached files.

Reviewer #2: No

---

## [Author Response · Author response to Decision Letter 1]

2 Jul 2023

29/6/23

Additional Editor Comments:

The authors have done a good job with the revisions. For future manuscripts, I would encourage you not to write vague things in your responses letter like “This has been changed in the Discussion”. It is far easier for referees, if you cut and paste in your revised text in the cover letter, so they can quickly see what you have done, rather than having to hint through the manuscript for the changes.

Thank you for highlighting this. ‘Response to reviewer’ document has been changed to show the changes made.

For the histogram figures, I would perhaps not go with subtle differences in shades of blue. This is not very clear. Also note on the y-axis title, “Proportion” means a number from 0 to 1. “Percentage” means a number from 0-100. Also I would give a more information axis title that just “Proportion”. The normal approach when producing a scientific graph is to have an axis title and, in brackets, the unit. “Percentage” is the unit.

Well done. Graeme Hays

The colours used in the bar charts have been changed to make these easier to read. Additionally, the y-axis labels have been updated to more descriptive and accurate.

Previous Comments

Formatting edited to meet requirements. Font sizes of headings and subheading updated. File names changed. 

Minimum dataset included with this draft.

Minimum dataset included with this draft.

Additional Editor Comments:

We now have three detailed and constructive reviews of your manuscript. You will see that the referees liked aspects of the manuscript but there are several concerns. You’ll see that the referees consistently thought that your writing was a little too parochial and the results poorly presented. On balance I think that if you take care with a major revision, this manuscript might still be suitable for PLoS1 and so I am recommending that you revise the manuscript taking all the comments into consideration. I expect that this major revision will take some time as all the referees called for some substantive changes.

I look forward to seeing a revision.

Reviewers' comments:

Reviewer's Responses to Questions

Comments to the Author

1. Is the manuscript technically sound, and do the data support the conclusions?

Reviewer #1: No

Reviewer #2: Yes

Reviewer #3: Partly

Data collection was performed as rigorously as possible over the time period.

2. Has the statistical analysis been performed appropriately and rigorously?

Reviewer #1: No

Reviewer #2: Yes

Reviewer #3: No

Statistical analysis kept relatively simple and based on observations only.

3. Have the authors made all data underlying the findings in their manuscript fully available?

Reviewer #1: No

Reviewer #2: No

Reviewer #3: No

Dataset included in this draft.

4. Is the manuscript presented in an intelligible fashion and written in standard English?

Reviewer #1: No

Reviewer #2: Yes

Reviewer #3: Yes

Manuscript edited based on suggested grammatical revisions. Proofread for further grammar and typographical errors.

5. Review Comments to the Author

Reviewer #1: This study presents an interesting insight on the prevalence of ghost-fishing in the Maldives, driving strandings of sea turtles. However, at present, this work is not placed in sufficient context of the wider literature, limiting the interest of the readership. 

Introduction refocused to broaden interest. Discussion is already in context of wider literature and discusses peer reviewed studies of sea turtle injuries and mortalities in a global context, as well as other species and threats from within the Maldives.

Also, no figures/tables were provided, making interpretation very difficult.

Figures included to clarify results.

I have made detailed recommendations up to the end of the results; however, I have not assessed the discussion, due to the extent of revision required throughout. With careful and detailed revision, this could make a very interesting contribution to the journal.

Abstract

Line 25-7; this is too broad; what is the key knowledge gap that needs to be explored in general? Maybe amend to something like “Quantifying how ghost fishing contributes to anthropogenic-driven losses of sea turtles could help guide marine management; however…” why has this not been done so far, what limitations have stopped this, why do you have the tools to achieve it now?

Reworded to ‘Quantifying the effect of human actions on these threatened species can help guide management strategies to reduce adverse impacts.’

Line 27-8, this is melodramatic and again too broad; what information is missing in this region, and why, due to a lack of funds, equipment, technology? Clarify.

Clarified to ‘such assessments require extensive effort and resources and as such have not been carried out in many areas of important sea turtle habitat’

Line 29 – separate your aim from your key finding

“Here, we utilised 12 years of stranding and rehabilitation data from the Republic of the Maldives to identify key threats to marine turtles in this region.”

Reformatted to ‘Here, we utilise 12 years of stranding and rehabilitation data from the Maldives to identify key threats to marine turtles in this region. Olive ridley turtles were found stranded or injured most frequently…’

State which 12 years and whether this coverage was nationwide or biased.

‘Here, we utilise 12 years (2010-2022) of stranding and rehabilitation data from the Maldives’. It was felt that including coverage was not appropriate here. However, this is highlighted in the results ‘from 18 different atolls with the central atolls (Male, Baa, Ari) over-represented’.

Then state the results

“Entanglement in ghost nets represented the leading cause of injuries and strandings (66.2% of cases).”

Reworded to ‘Anthropogenic factors were the primary cause of injury or stranding in 75.2% of cases with entanglement in ghost fishing gear being the most common (66.2% of all cases).’

Line 33. Make this a new sentence, and avoid referring to the wider literature.

It is not clear what the “regions” you were referring to here, if you want to make a comparison to the wider literature, this should be done in the discussion, as here it would mean your needing to add citations, which is not permitted. The Abstract should present your results.

This phrase has been removed

Line 35, what do you mean by “non resident” and “resident”– non resident to where, as you state the study was done at a national scale? This does not have context. The reader must be informed of why you are using these different terms. This should be your first statement of your Abstract results, then percentage of human causes, then the bycatch vs ghost fishing.

Reference to residency removed. Section reworded to ‘Olive ridley turtles were found stranded or injured most frequently (74.7% of total cases), along with hawksbill (15.2%), and green (10.1%) turtles. Anthropogenic factors were the primary cause of injury or stranding in 75.2% of cases with entanglement in ghost fishing gear being the most common (66.2% of all cases). Other causes of morbidity, such as from turtles being kept as pets (5.6%), boat strikes (<1%), bycatch (<1%), and poaching (<1%) were recorded less frequently. Olive ridley turtles were more likely to have injuries associated with entanglement than other species with a peak in admissions during the northeast monsoon in the period following the known arribada nesting season in nearby India. However, turtles admitted to rehabilitation following entanglement were released a mean of 70 days sooner and had 27.5% lower mortality rates than for other causes of admission.’

Line 37-38, this result while interesting, does not contribute to your narrative here; remove it, or give some interpretation of why it is relevant to the ghost fishing bias.

Clarified as ‘peak in admissions during the northeast monsoon in the period following the known arribada nesting season in nearby India’

“and identifies several

40 key areas where current knowledge of sea turtle ecology in this region is lacking.” – delete, this does not convey useful information; it is a throwaway statement that could be placed on any paper.

Statement removed

“However, for the

41 findings of this study to impact the status of marine turtles they must be implemented into practice and

42 policy.” – also delete this, it weakens your final statement of the previous sentence.

Removed. End of abstract now reads ‘This study highlights the high prevalence of ghost net entanglement in sea turtles within the Maldives. The topic of ghost fishing is of global importance and international cooperation is critical in tackling this growing issue.’

Abstract reworded taking all comments into consideration

Keywords – missing

Keywords added: ‘Keywords: sea turtle, morbidity, mortality, ghost fishing, Indian Ocean, olive ridley, Maldives, conservation, rehabilitation, endangered species, ghost net’

Introduction

Please check that citations are formatted correctly.

One citation reformatted from Harvard to Vancouver style. (‘Wallace 2011’)

This entire section requires careful revision. The introduction is too long and too turtle centred. There are too many short paragraphs with no clear message. I advise that this section is carefully revised taking the broad readership of the journal into consideration.

Introduction rewritten based on below comments and suggestions

I would advise the following

Paragraph 1 – What is the current prevailing threat to marine vertebrates globally, turtles are one of several groups threatened by human activities at sea. There is a vast global literature on this, and you can identify ghost fishing as one such component here and how it compares to other human caused mortality. 

‘Human activities have substantial impacts on the worlds’ oceans and the species which live in them (1). Anthropogenic factors such as overexploitation, habitat loss, climate change, invasive species, disease, and pollution can negatively affect wildlife populations and contribute to species declines or extinctions (1,2). The impacts of these activities are more pronounced in large bodied species which are subject to more intense pressures e.g. from overexploitation, and are vulnerable to extinction through slow life histories (3,4). As marine megafauna convey a variety of environmental, economic, cultural and social benefits disproportionate to the overall percentage of species they represent, and additionally can act as umbrella species, they should be a priority for conservation efforts (3,5).’

Paragraph 2 – Ghost fishing – focus on detail on this, and whether threats are uniform globally across marine vertebrates, and if not identify potential reasons why not.

‘Marine turtles are one group of marine megafauna under threat of extinction through human activities; primarily through interactions with the fishing industry, overexploitation, and marine pollution (6–8). Although six of the seven species of marine turtle are considered to be under threat of extinction, through extensive conservation efforts populations are stable or increasing in many regions (9–11). However, as the type and magnitude of threats to marine turtles varies between geographic regions, it is important to consider that management decisions to mitigate anthropogenic impacts in one location may not be effective in another (12–14). To safeguard against future losses and facilitate further population recovery it is critical to identify and quantify threats to marine turtles on a regional scale.’

Paragraph 3 – Sea turtles and threats at sea; draw on the wide literature of human caused threats at sea (see Wallace et al. 2010 and associated publications), and in particular explore the literature around ghost fishing and sea turtles; if it is limited, here, highlight why, what approaches work best with detecting this effect, i.e. draw on rehabilitation and strandings and the pros/cons of using such literature.

Regardless of the importance of assessing threats to marine turtles, the process remains challenging: all species are elusive with pelagic life stages, making gathering the large datasets required for accurate evaluations labour intensive and costly (15). Given the extensive resources required to assess threats these have been performed only in certain well-studied populations; namely of green turtles in the Americas and Australia, and loggerhead turtles in the Mediterranean (8,12,16–18). However, threats to marine turtles remain unassessed in many regions, including areas with significant populations (15).

The Republic of the Maldives (Maldives) is one region of important marine turtle habitat where a comprehensive evaluation of threats has not been carried out (11). However, a rapidly expanding and increasingly environmentally focused tourism industry over the past few decades has facilitated the collection of comprehensive and long-term data across many areas of marine science in the Maldives. (19,20). Here, we utilise data collected from stranded turtles and those admitted into rehabilitation centres within the Maldives to evaluate the threats to marine turtles in this region.

Stranding data is a common method of assessing causes of morbidity and mortality in marine turtles (8,12,21). Although stranded turtles found on beaches or floating on the ocean’s surface only represent a small proportion of total deaths and injuries; strandings are considered representative of threats and allow estimations of the scale of local hazards to be made (14,21). Where stranded individuals are found alive and admitted to rehabilitation centres, longitudinal observations made on progress and recovery can provide additional data towards a more comprehensive overview of threats to marine turtle populations in a region (22). 

Paragraph 4 – your study aims and hypotheses, along with main approach, study region. What did you expect to find?

Five of the seven globally recognised species of marine turtle have been recorded in the Maldives. Green (Chelonia mydas) and hawksbill (Eretmochelys imbricata) turtles are permanent residents and are sighted frequently throughout the region (23). Both species hold neritic foraging grounds which are established after an initial pelagic life-stage as young juveniles. Nesting is reported in several atolls with animals known to migrate from the Chagos archipelago; indicating that the Maldives provides important nesting habitat for turtles in the region (24–27). A recent regional IUCN evaluation has classified hawksbill turtles as ‘critically endangered’ and green turtles as ‘endangered’, closely matching global assessments, although assessments by other parties suggest that populations in the area are stable (11,28).

Olive ridley turtles are found more frequently in pelagic habitats than neritic and are known to have large nesting populations along the east coast of India (29). In the Maldives they are most frequently sighted offshore and have no known resident or nesting populations (23). However, olive ridley are found entangled in ghost nets, defined as fishing nets which have been lost or discarded, with relatively high frequency within the atolls, particularly during the northeast monsoon (January to March) where mass nesting (also known as arribada behaviour) along the east coast of India overlaps with a peak in trawl fishing in the same area (30–32). The strong monsoon currents then wash injured and tangled turtles into the Maldives (30–32).

Although both loggerhead and leatherback turtles have been reported within the Maldives, both species are infrequent passing visitors with no known resident populations (23). 

This study represents the first long-term, multi-species analysis of sea turtle morbidities and mortalities in this area of the Indian Ocean. Using 12 years of stranding and rehabilitation data collected within the Maldives we aim to: analyse initial status and cause of injury in stranded animals, compare these between species and life stages, determine overall mortality rate of animals found alive, and identify seasonal patterns in strandings.

Introduction rewritten based on the above suggestion to broaden the context

Line 102 – “by comparing these findings to other regions” – this statement needs careful reconsideration; this implies you conducted an exhaustive literature review; if this is the case, then in your Paragraph 4, where you state your aims, you also need to state that your work is a combination of both empirical data and a wider literature review. This must be clarified in the abstract and the exact approach of the review stated in the methods.

Phrase removed

Line 104-5, this is fine

Line 106 – it is not clear how this is feasible, remove

Removed

Methods

Line 110, just put the coordinates in parentheses at the end of the last sentence.

Revised: ‘…axis around 400km to the southwest of India (07°06’N - 00°41’S, 72°32’E - 73°45’E) (Fig 1).’

A figure is required showing the entire nation and position of the two atolls, along with the catchment area for strandings assessed here.

No tables or figures are provided on the manuscript, making evaluation difficult.

Figures included (map of the location of the Maldives including rehab centres and stranding sites, summaries of admissions and results).

Line 128 – what are circumstantial data, give some examples of what this includes.

Clarified in text. E.g. Date, location. Rephrased as ‘Data was recorded from injured or stranded turtles which were reported to the MSTCP between March 2010 and September 2022. Animal handling and husbandry practices during this process followed recommended and best practice sea turtle care and management guidelines (37–40). Turtles reported to the MSTCP had a standard set of information recorded: species, curved carapace length (CCL), life stage, sex, date found, identity of the reporting party, initial disposition, cause of injury or mortality, details of injuries or abnormalities present, final outcome, and date of final outcome. Life stages were categorised’

Line 144 – at the site or on arrival? If at the site, how was this standardized across different people finding animals?

Measured on arrival to rehabilitation or on site wherever possible. CCL was left as NA if not measured. Status was recorded on arrival. If the animal was released immediately status was based on the discoverers' description. Clarified in text as ‘This information, excluding data pertaining to the final outcome of the case, was recorded on admission for rehabilitation. In cases where this did not occur data was recorded from verbal descriptions and visual media (photos and videos) provided by the discoverers of the turtle.’

Change disposition to status throughout.

Changed from ‘disposition’ to ‘status’ throughout.

Line 162, were necroposies done to check for blockages? If not, you cannot state if this is natural.

Necropsies performed wherever possible. However, cases where this was not possible had cause of death recorded as ‘unknown’. ‘Cause of injury or mortality was determined by several means: initial status, clinical examination and post-mortem examination of deceased individuals.’

Line 163, it is not clear why husbandry fits here. At stranding, husbandry is not relevant; if they survive rehabilitation cannot really be determined as a husbandry issue. The context of husbandry needs careful consideration. How do you assess if husbandry was poor?

Clarified in text. This point refers to cases of turtles being kept as pets as injuries or illness associated with poor husbandry resulted in admission to rehabilitation. ‘…whilst anthropogenic causes of injury incorporated entanglement, boat strikes, hook injuries,and, for animals kept as pets, poor husbandry.’

You do not state anywhere what area your strandings covered – is it national? How do you overcome bias to populated areas?

Clarified in text. Stranding reports are national but show bias towards the central, more densely populated atolls. ‘In total, 459 turtles were reported as stranded or injured between 2010 and 2022 from 18 different atolls with the central, more densely populated atolls (Male, Baa, Ari) over-represented’

Results

Supporting tables and figures are required.

Included

This needs to be broken into 3-4 key subsections each supported by a clear figure. Shift all percentages into tables, then revise your entire text to focus on the key points of interest.

By delineating clear subsections, you can then develop some clear hypothesis testing.

Results broken into subsections with separate headings. Figures/tables included but numbers/percentages left in-text to support written results.

Line 185, how do you get a female bias for adult hawksbills if no adults were recorded?

Error remedied: ‘The adult sex ratio, in turtles where this was recorded, was female biased in olive ridley and green turtles with a female: male ratio of 2.75:1 and 2:1 respectively. As no adult hawksbill turtles were recorded it was not possible to determine an adult sex ratio for this species (Fig 3a).’

Line 185, you simply state, The adult sex ratio was X:X for green turtles and X:X for olives. What is this sex ratio based on, the stranded animals? If something else, more explanation of how it was obtained is required in the methods.

Clarified in text. Ratio refers to all stranded individuals for which sex was recorded (see above).

Line 188-191, what about inhabited vs uninhabited islands?

See figures for distribution of strandings across atolls. Highlighting inhabited/uninhabited islands is not practical as land area is very small comparative to total area covered in this study. Even at the level of individual atolls land masses are barely visible.

A map showing the distribution of all strandigns for the 3 species (different colours) and size classes should be presented, along with highlighting inhabited islands.

Map of stranding sites included in figures. Including sex of the individual made the figure too complicated. Highlighting inhabited/uninhabited islands not practical as islands are too small to be visible on a national, or even an atoll scale.

Line 192, while turtles were found entangled in nets, this might easily be just the end point, i.e. turtle gets infection, floats, becomes entangled; entanglement itself might not be the primary cause.

How do you address this? For instance, all animals could have been floating, then become entangled; the key point is to find the initial cause for floating.

Here we are reporting the initial condition in which animals were found. It is felt that discussing the various causes of buoyancy syndromes in marine turtles is out-with the scope of this paper. However, cataloguing this and other injuries in more detail is currently an area of ongoing work within this dataset.

Line 192-5, this needs to be presented graphically.

Graphs are required to support the data throughout this section; once you have graphs, you can then interpret the key points of interest, rather than listing everything.

Graphs of key results included. However, as results are broad, including figures of all results would result in a high number of figures being included.

Line 219 – were necropsies/xrays conducted?

Necropsies were carried out where possible to determine cause of death. Cases were determined to have an ‘unknown’ cause where necropsies were unable to be performed or where the necropsy results were unconclusive. This has been clarified in the methods section (see above). X-rays were performed in very rare cases only as this involved transporting patients to a human hospital. Until very recently no veterinary x-ray units were available in Maldives. 

Reviewer #2: Here the causes of injury and reasons for mortality are reported for turtles found stranded or floating in the Maldives, Indian Ocean. Is it concluded that entanglement in ghost nets is a key source of injury and mortality.

This is an interesting manuscript that I enjoyed reading. Some nice data are presented. I have some suggestions to improve the final version. With a little care, these revisions should be straightforward to deal with.

1. Line 25. “Anthropogenic activities are driving the global decline of sea turtle populations.

Sensationalist as well as incorrect. You need to be a little more honest.

See:

Mazaris AD et al. (2017). Global sea turtle conservation successes. Science Advances 3: e1600730. https://doi.org/10.1126/sciadv.1600730

Rephrased to ‘Anthropogenic activities can negatively affect sea turtle populations.’

I think throughout (e.g. start of Intro, lines 45-52) you need to say that at many sites around the world, sea turtle numbers are increasing due to conservation efforts (Mazaris). But regardless, sea turtles still face threats and so to facilitate further population increases it is important to both identify and then mitigate threats.

Updated in text to ‘Although six of the seven species of marine turtle are considered to be under threat of extinction, through extensive conservation efforts populations are stable or increasing in many regions (9–11). However, as the type and magnitude of threats to marine turtles varies between geographic regions, it is important to consider that management decisions to mitigate anthropogenic impacts in one location may not be effective in another (12–14).’

2. Line 34. “ … causes of admission …”

Reads poorly as, in the abstract, you have not mentioned anything about admission to a rehab centre being the basis for the data collection.

Reworded to ‘Here, we utilise 12 years of data (2010-2022) collected from marine turtle strandings and rehabilitation cases from across the Maldives to identify the key threats in this region’

3. Line 77. “Despite having large populations of sea turtles, the central Indian Ocean is one region where evaluation of threats has been neglected …”

Can you add some details about “large populations”. E.g. see work of Jeanne Mortimer and others for regional assessments of nesting numbers – see: https://doi.org/10.1017/S0030605319001108

Clarified in text to ‘The Republic of the Maldives (Maldives) is one region of important marine turtle habitat where a comprehensive evaluation of threats has not been carried out (11).’ However, there is currently no data published on the size of turtle populations for this region.

4. Line 85. “However, numbers of both species are currently declining.”

Can you qualify this statement ? Where is this info published ?

Since the first submission of this paper Stelfox et al have published evidence that marine turtle populations are stable. This has been updated in the text. ‘A recent regional IUCN evaluation has classified hawksbill turtles as ‘critically endangered’ and green turtles as ‘endangered’, closely matching global assessments, although assessments by other parties suggest that populations in the area are stable (11,28).’

5. Line 138. “Green turtles with CCL greater than 95cm (males) and 100cm (females) were classed as adults …”

Can you estimate the probability of a mis-id of life stage, e.g. a sub-adult male being classified as a female ? This is important as later you report sex ratios, but I suspect you’ll have quite a few mis-ids.

This is addressed in the discussion. Anecdotal evidence suggests that hawksbill and green turtles in this region are smaller than neighbouring populations. However, there is currently no data to back this up. ‘Here, we show the overall demographics of stranded turtles broadly concur with findings from other global regions. Juvenile animals are recorded more frequently than adults and, within the group of adult animals whose sex is determined, females are more abundant than males. These trends are common to strandings and in-water observations of marine turtles in multiple regions, including the Maldives, and represent overall population demographics (27,31,44). Although a predominance of juveniles within turtle populations is a common finding, our study recorded no adult hawksbill turtles. Size of adult turtles is known to vary between regions, and it has been suggested that both green and hawksbill adults may be smaller in the Maldives than in surrounding regions (23). As no definitive measurements of nesting adults are available for this region, utilising measurements taken in other regions, as here, may result in an underestimation of adults. However, proportions of adult green turtles here are comparable to a large dataset collected from Mexico, suggesting that the adult: juvenile ratio for this species may be representative (13). Moreover, further work is required to confirm this observation’

6. Lines 162. “… cachexia (emaciation with no discernible primary cause …”

How was this recorded ? e.g. some threshold residual value from a length:weight relationship ? Or some objective scoring (e.g. see Heithaus/ Thomson work in Shark Bay) or did you just make a subjective best guess ? That’s fine. Just explain what you have done.

Body condition index calculated using method published by Norton and Wyneken (2015)- clarified in text to ‘cachexia (emaciation with no discernible primary cause as determined by Body Condition Index (BCI)’

7. Line 207. “… was entanglement in ghost fishing nets or similar materials (66.2%, n=251) with wounds characteristic of net or line entrapment identified in both entangled turtles (n=211) and in those found floating or beached (n=40) …”

If a turtle just has wounds characteristic of nets or lines, how do you know this was a “ghost net” and not just a net being used by a fisherman who released the turtle ?

Injuries origin was determined using previously defined criteria (eg. Archibald et al. 2018). The characteristic lacerations caused by extended entanglement would typically not occur with a turtle being line caught; hook injuries to the mouth being more common in these cases. Additionally, in Maldives, methods of fishing which could conceivably cause such injuries such as net or longline fishing, are not permitted (see introduction).

8. Line 226. Boat strike. Was this associated with nearby resorts ?

Very interesting point. All 9 cases were reported by resorts but as resort and local islands exist in close proximity and travelling/drifting would have occurred it would not be possible to determine whether resort or local traffic was the cause of these cases.

I would have a few lines of the Discussion about boat strikes. This has been widely reported around the world (e.g. see Gail Schofield’s work in Greece, https://doi.org/10.1002/ecy.3027 and also in the USA, see Tony Tucker). Boat strike is particularly associated with high boating traffic and had led to speed restrictions in high use turtle areas in the US and Greece.

Short paragraph on boat strikes added to the discussion. ‘Boat strikes are a commonly noted cause of anthropogenic injuries in marine turtles, particularly in areas with high water-based traffic (8,12,65). Here we find boat strikes account for 2.4% of injuries, a figure comparable to other regions (8,12). The not-insignificant risk posed by boat strikes to sea turtles has led to restrictions of water-traffic movements in areas of high turtle activity in several countries (66,67). Although similar policies are currently implemented in the Maldives for other megafauna species, eg. manta rays and whale sharks (68), currently no such delineations exist for turtles. Further work is required in this regard to identify areas with high turtle populations and establish the risk presented by boat strikes within these zones.’

9. Line 286. You mention where Olive Ridley likely may have come from. Similarly can you detail some of the areas that adult greens and hawksbills have come from ? There has been satellite tracking showing adult green from the Chagos Archipelago travel to the Maldives. But adult hawksbills from Chagos likely do not travel to the Maldives … rather all the tracked animals stay within Chagos (e.g. see https://doi.org/10.1098/rsif.2021.0859). So I guess any adult hawksbills you see also nest in the Maldives ? We suspect that the majority of green and hawksbill cases presented to us are resident animals with a few originating from places like Chagos. However, there is currently no data to support this that I am aware of outside of the Chagos post-nesting tracking data.

10 There are no figures. Perhaps think of 1 or 2 key results to show in a graph ?

e.g. a map showing the number of reports came from different places in the Maldives. Perhaps a photo of a stranded turtle in a ghost net. Noted and included

11. “Poaching”. I could not see anything in the methods about how this was ascertained. Describe your method. Include a photo ?

Clarified in methods. ‘The discovery of an entire carapace or plastron with toolmarks was considered to be indicative of poaching.’

In summary, a nice piece of work with some interesting results. Well done to the authors on completing such a nice study.

Thank you!

Reviewer #3: See attached review with suggestions to improve the manuscript.

It was difficult to state that the analysis was appropriate when no supporting data were provided and no tables or figures were included

Figures, tables and data now included.

This sea turtle study presents 12 years of stranding and rehabilitation data from the Maldives to report key threats to sea turtles, identifying entanglement in ghost nets as one of the key anthropogenic causes of injuries. The dataset is considerable and includes 459 turtles. Clear study objectives are presented including variation in cause of injury between species and life stages, estimation of mortality rate and identification of seasonality in strandings. Providing that the review comments below are addressed, the study would advance our understanding of some of the underreported anthropogenic impacts from fishing, as already highlighted by a number of studies in the region (e.g., Stelfox et al) cited in the manuscript.

The findings are clearly presented but the manuscript could be much improved by re-arranging content into subsections with relevant subtitles in Methods and Results. 

Methods and results re-arranged. Appropriate subheadings included

No figures and/or tables were included so it was difficult to interpret results. Additionally, no underlying data were provided (that I could access) so it was difficult to comment on the review question ‘Have the authors made all data underlying the findings in their manuscript fully available?’. To this end, a supplementary table providing summary data would be helpful to support the growing body of literature on this topic in addition to providing a link to the Reefscapers website.

Figures and tables added along with data

Some specific comments:

Abstract: 

State number of atolls and relative location where data were collected 

Stated in results and included in Fig 1. ‘In total, 459 turtles were reported as stranded or injured between 2010 and 2022 from 18 different atolls with the central, more densely populated atolls (Male, Baa, Ari) over-represented (Fig 1).’

The study is for one small area of the Indian Ocean and I suggest re-phrasing ‘high prevalence of ghost net entanglement in the Indian Ocean’. If a review has shown high prevalence elsewhere in the IO then include the results.

Noted and phrase removed from text.

Introduction:

I recommend to commence with a paragraph describing different threats to turtles (and introducing the various anthropogenic threats) before focusing on species and assessment.

L77: define what you are including in the term Central Indian Ocean to help with readers’ understanding as typically this region is split between NWIO and SWIO and NEIO with respect to sea turtle regional management units (RMUs) Wallace et al. 2010. http://www.doi.org/10.1371/journal.pone.0015465

Reworded/rephrased throughout for clarity and to fit with accepted nomenclature.

L79: insert the citation as numeric.

Reformatted.

L84: provide more context about population connectivity in the Western Indian Ocean, for example a number of green turtles have migrated to Maldives from Chagos Archipelago in the south (Hays et al. 2020 https://doi.org/10.1016/j.cub.2020.05.086) which provides critical nesting habitat for turtles from around the region (see Mortimer et al. 2020 http://www.doi.org/10.1017/S0030605319001108). 

Included in introduction: ‘Green (Chelonia mydas) and hawksbill (Eretmochelys imbricata) turtles are permanent residents and sighted frequently throughout the region (23). Both species hold neritic foraging grounds which are established after an initial pelagic life-stage as young juveniles. Nesting is reported in several atolls with animals known to migrate from the Chagos archipelago; indicating that the Maldives provides important nesting habitat for turtles in the region (24–27).. A recent regional IUCN evaluation has classified hawksbill turtles as ‘critically endangered’ and green turtles as ‘endangered’, closely matching global assessments, although assessments by other parties suggest that populations in the area are stable (11,28).’

Results:

I would suggest to the authors to provide a number of graphs to show key results, for example: 

To support the statements about turtles stranded or bycaught turtles in the Maldives : provide a map with indication of numbers of each case category at each location

To support statements about seasonal patterns of admissions : provide a graph comparing number of cases per month of the year

To support statements about entanglement being key cause of injuries or death: graph reporting number of cases for each threat identified

Provide a graphical overview of relative proportion of life stages and species included in the study

Figures created and included to illustrate key results.

Discussion:

This is quite lengthy and could be improved by re-ordering to commence the section with key results that are discussed in context of the literature, followed by several sections with appropriate sub-titles to discuss the various findings. 

Noted and trialled. The discussion opens with the key finding of the high prevalence of ghost net entanglement. Sub-titles also trialled but it was felt that this broke the discussion into too many short sections.

There are some statements that should only be included if supported by citations (e.g. L418 turtle populations in the Central IO are some of the most threatened in the world …)

Rephrased or cited throughout discussion.

It is worthwhile considering potential bias in the dataset due to method of entanglement. Relative contributions of ghost nets to entanglement and injury/mortality of turtles is possibly overreported compared to other causes of anthropogenic threat. This is because ghost nets are likely to come ashore as they are usually buoyant and carried in surface waters. Turtles that snag on fishing line caught on the reefs are less likely to be included in the dataset. Consider other anthropogenic factors in your discussion, e.g. those reported by Casale et al. 2010 https://doi.org/10.1002/aqc.1133

Point well made; discussion updated to address this. ‘Although it is important to note that entanglement cases are likely to be over-represented in this study as the often-buoyant ghost nets will increase the likelihood of affected individuals being recovered, it should be considered that this bias will equally affect all other regions. The comparative scale of entanglement cases identified here should not be underestimated.’

L294: As suggested for the introduction, a definition of the area (Central Indian Ocean) and a map of the study site would help support this statement. 

Rephrased throughout.

---

## [Editor Report · Decision Letter 2]

13 Jul 2023

Evaluation of sea turtle morbidity and mortality within the Indian Ocean from 12 years of data shows high prevalence of ghost net entanglement.

PONE-D-22-34145R2

Dear Dr. Himpson,

We’re pleased to inform you that your manuscript has been judged scientifically suitable for publication and will be formally accepted for publication once it meets all outstanding technical requirements.

Kind regards,

Graeme Hays

Academic Editor

PLOS ONE

Additional Editor Comments (optional):

The authors have made some effort to revise the manuscript in line with the final comments. I think this manuscript can now be accepted for publication. Graeme Hays
---

## [Editor Report · Acceptance letter]

17 Jul 2023

PONE-D-22-34145R2 

Evaluation of sea turtle morbidity and mortality within the Indian Ocean from 12 years of data shows high prevalence of ghost net entanglement. 

Dear Dr. Himpson:

I'm pleased to inform you that your manuscript has been deemed suitable for publication in PLOS ONE. Congratulations! Your manuscript is now with our production department. 

Kind regards, 

on behalf of

Professor Graeme Hays 

Academic Editor

PLOS ONE